Lucia S. Layritz, Konstantin Gregor, Andreas Krause, Stefan Kruse, Ben Meyer, Thomas A. M. Pugh, Anja Rammig

# Disentangling future effects of climate change and forest disturbance on vegetation composition and land-surface properties of the boreal forest

Lucia S. Layritz[1], Konstantin Gregor[1], Andreas Krause[1], Stefan Kruse[2], Benjamin F. Meyer[1], Thomas A. M. Pugh[3, 4, 5], and Anja Rammig [1]

[1]Land Surface–Atmosphere Interactions, TUM School of Life Sciences, Technical University of Munich, Freising, Germany
[2]Alfred Wegener Institute Helmholtz Centre for Polar and Marine Research, 14473 Potsdam, Germany
[3]School of Geography, Earth and Environmental Sciences, University of Birmingham, Birmingham, B15 2TT, United Kingdom
[4]Birmingham Institute of Forest Research, University of Birmingham, Birmingham, B15 2TT, United Kingdom
[5]Department of Physical Geography and Ecosystem Science, Lund University, Lund, Sweden

**Correspondence:** Lucia S. Layritz (lucia.layritz@tum.de)

**Abstract.** Forest disturbances can cause shifts in boreal vegetation cover from predominantly evergreen to deciduous trees or non-forest dominance. This, in turn, impacts land surface properties and, potentially, regional climate. Accurately considering such shifts in future projections of vegetation dynamics under climate change is crucial but hindered e.g. uncertainties in future disturbance regimes. In this study, we investigate how sensitive future projections of boreal forest dynamics are to additional changes in disturbance regimes. We use the dynamic vegetation model LPJ-GUESS to investigate and disentangle the impacts of climate change and intensifying disturbance regimes in future projections of boreal vegetation cover as well as changes in land surface properties such as albedo and evapotranspiration. Our simulations find that warming alone drives shifts towards more densely forested landscapes, and more intense disturbances reduce tree cover in favor of shrubs and grasses, while the interaction between climate and disturbances leads to an expansion of deciduous trees. Our results additionally indicate that warming decreases albedo and increases evapotranspiration, while more intense disturbances have the opposite effect, potentially offsetting climate impacts. Warming and disturbances are thus comparably important agents of change in boreal forests. Our findings highlight future disturbance regimes as a key source of model uncertainty and underscore the necessity of accounting for disturbances-induced effects on vegetation composition and land surface-atmosphere feedback.

## 1 Introduction

Climate change induces widespread changes in ecosystem state and functions (McDowell et al. (2020); Allen et al. (2010)). Next to changing mean conditions, climate change is expected to lead to an increase in extreme climatic events in many regions (Calvin et al. (2023); Field (2012); Rahmstorf and Coumou (2011)). On the ecosystem level, this alters regimes of ecosystem disturbances such as fire, windthrow, or biotic agents (McDowell et al. (2020); Seidl et al. (2017); Reichstein et al. (2013)).

Such disturbances may lead to widespread tree mortality and loss of forest cover. While they are inherent elements of many forest ecosystems, changes in disturbance regimes (that is, changes in their frequency, intensity, or size) can nevertheless have profound impacts, especially in systems already subjected to environmental pressure, where resilience and regeneration abilities are low (Pugh et al. (2019a); Allen et al. (2010); Johnstone et al. (2016)). Understanding the impact of disturbances is therefore crucial for reliably projecting the future development of forest ecosystems under climate change.

The boreal forest, or Taiga, is the second largest terrestrial biome in both area and carbon mass (Pan et al. (2011); Malhi et al. (1999)), spanning the Northern Hemisphere in a circumpolar band between approximately $50^{o}N$ and $70^{o}N$. (Pfadenhauer and Klötzli (2020)). Its characteristic vegetation are conifer forests, dominated by various species of *Abies*, *Picea*, and *Pinus* in North America and Western Eurasia and by *Larix* in Siberia (Pfadenhauer and Klötzli (2020)). In the boreal forest, disturbances are an integral part of ecosystem dynamics, and tree species are thus adapted and resilient to historical disturbance regimes

(Pfadenhauer and Klötzli (2020); Ilisson and Chen (2009); Johnstone et al. (2010)). However, evidence from both paleoecology (Peros et al. (2008); Edwards et al. (2005)) and recent field surveys (Baltzer et al. (2021); Mack et al. (2021); Johnstone et al. (2010); Brice et al. (2020)) suggest that changing disturbance regimes can disrupt existing successional cycles and induce reorganization of the complete ecosystem. In many places, evergreen needleleaf trees fail to regenerate after disturbance and transitions to broadleaf summergreen forests or non-forest ecosystems are reported (Baltzer et al. (2021); Mack et al. (2021)).

Boreal forests influence regional and global climate (Bonan (2008); Randerson et al. (2006)). Aside from being a carbon sink, vegetation composition influences surface properties such as albedo or evapotranspiration. Swann et al. (2010); Liu et al. (2019); Rogers et al. (2013); Bonan (2008); Chapin et al. (2005)). Shifts in vegetation composition can, therefore, result in significant alterations to the carbon, water, and energy balance of the region (Mack et al. (2021); Boisier et al. (2013); Zhang et al. (2013); Alexander et al. (2012); Swann et al. (2010); Bonan (2008)). Consequently, understanding the role of

disturbance-induced vegetation shifts in future vegetation and climate dynamics and their accurate representation in climate change projections are important.

Several studies have tackled part of this question: Kim et al. (2024); Wang et al. (2020) and Sulla-Menashe et al. (2018) have aimed to quantify the disturbance effect from observations. However, this remains incomplete as disturbed sites are also subject to background climate change, and the disturbance effect can therefore not be isolated. Additionally, observational time series

of forest dynamics, especially large-scale assessments from remote sensing, are still relatively short ($\sim$ 30 years) compared to the multidecadal to centennial time scales of forest succession. While longer time-series of post-disturbance dynamics can be constructed from chronosequences, those rely on a space-for-time substitution effect and do not take temporal changes in climate into account. Therefore, it remains difficult to pinpoint if the observed changes will be permanent or transient in nature.

Process-based vegetation models are a prime research tool to complement observational findings in these regards, as they

allow for the factorial simulation of different drivers over longer time periods. Zhang et al. (2020); Wårlind et al. (2014); Zhang et al. (2013); Warszawski et al. (2013); Foster et al. (2019), and Wolf et al. (2008) explored the impact of future climate change on boreal forest dynamics and land surface properties with dynamic vegetation models (DVMs) without considering changes in disturbance regimes. Rogers et al. (2013) investigated the impact of changing disturbance regimes on land surface properties and climate but did not consider climate change effects on vegetation. Hansen et al. (2021); Brice et al. (2020); Foster et al.

(2022) and Mekonnen et al. (2019) explored the interplay of climate change and disturbance regimes for parts of Alaska and Canada over the 21st century. None of these studies explored long-term effects beyond the 21st century and regeneration after disturbance. To our knowledge, there so far exist no biome-wide modeling studies that systematically investigate both 21st century and long-term future impacts of changing disturbance regimes, climate change, and their interaction for different climate futures.

In this study, we use the DVM LPJ-GUESS (Smith et al. (2001, 2014)) to fill this gap. LPJ-GUESS includes a process-based representation of plant physiology (photosynthesis, respiration, and evapotranspiration) and resolves vegetation structure on the level of individual tree cohorts, organized in multiple patches across the landscape. This allows for the simulation of disturbances, mortality, and establishment in a way well suited to study disturbance regimes and post-disturbance regeneration. We perform factorial simulation experiments of both different climate scenarios and external disturbance regimes spanning return intervals from the low end of what is historically observed to the high end of what is historically observed and projected for the future to disentangle their respective future roles in vegetation dynamics of high-latitudinal forests.

We address the following research questions: (1) What is the impact of climate change, changing disturbance regimes, and their interactive effect on boreal vegetation composition? (2) How do changes in vegetation influence climate-relevant biogeophysical land surface properties, namely albedo and evapotranspiration? (3) Are disturbance-induced changes permanent, or is vegetation able to regenerate once disturbance pressure is again lifted?

## 2   Methods

### 2.1   Vegetation modeling

#### 2.1.1   General LPJ-GUESS Model Description

We work with version 4.1 of the well-established dynamic vegetation model LPJ-GUESS (Smith et al. (2001, 2014); Nord et al. (2021)), driven by gridded temperature, precipitation, and downwelling shortwave radiation, as well as global $CO_2$ concentrations and soil properties. We here use a version parametrized for Arctic vegetation as summarized in Table A1. This version has been validated in previous studies, e.g. against species distributions maps (Zhang et al. (2013); Miller and Smith (2012); Wolf et al. (2008)), flux measurements of carbon cycle dynamics (Rawlins et al. (2015); McGuire et al. (2012), observational data of snow pack and soil properties (Pongracz et al. (2021); Chaudhary et al. (2020), as well as remotely-sensed land surface properties (Zhang et al. (2018).

Plant physiological processes follow the approach of LPJ-DGVM (Sitch et al. (2003)). As the focus of this study is on vegetation dynamics, we here briefly sketch out the main processes - described in detail by Smith et al. (2001) and Smith et al. (2014) - followed by a detailed description of population dynamics and disturbances. $CO_2$ and water fluxes are calculated by a coupled photosynthesis and stomatal conductance scheme based on the approach of BIOME3 (Haxeltine and Prentice (1996)). Each simulation year, the NPP accrued by an average individual plant is allocated to leaves, fine roots, and, for woody plants, sapwood, following a set of prescribed allometric relationships (Sitch et al. (2003)). Litter from phenological turnover,

mortality, and disturbances enters the soil decomposition cycle. For details on soil processes, including C-N dynamics and soil hydrology, refer to Smith et al. (2014); Sitch et al. (2003) and Gerten et al. (2004).

### 2.1.2 Population and disturbance dynamics in LPJ-GUESS

LPJ-GUESS employs a hierarchical model structure that allows for a detailed representation of population dynamics such as recruitment, competition and disturbance. Within each grid cell of climate data, multiple patches are simulated (25 patches are used in this study). Patches can be thought of as a random, independent sample of the gridcell, the model thus outputs the average across all patches in a grid cell. Vegetation dynamics within each patch emerge from growth and competition for light, space, and soil resources among woody plant cohorts and a herbaceous understory. Plants within a patch are represented by

different age cohorts of a number of plant functional types (PFTs) described by properties such as growth form, phenology, photosynthetic pathway (C3 or C4), and bioclimatic limits (see Table A2 for the PFT parametrization used in this study). Each age cohort includes multiple individuals of that PFT assumed to all have identical properties ('cohort mode'). Establishment and mortality are represented as stochastic processes on the cohort level. Both the probability of establishment and disturbance depend on the productivity of the PFT. The probability of mortality additionally depends on cohort age, and establishment on

the amount of light reaching the forest floor. PFTs are able to reproduce as soon as they are productive.

Disturbances in LPJ-GUESS are equally modeled as a stochastic process. The expected frequency of disturbances is encoded through the yearly disturbance probability $p_D$. If a patch experiences a disturbance, aboveground vegetation is converted to coarse woody debris and slowly decomposes over time. The patch structure here emulates heterogeneity in the landscape and accounts for the fact that disturbances are not necessarily stand-replacing on the landscape scale. We opted for this standardized

implementation of disturbances to reduce degrees of freedom in our experiments and be able to focus on downstream impacts.

### 2.2 Input Data

We run LPJ-GUESS with daily simulated climate from the Inter-Sectoral Impact Model Intercomparison Project (ISIMIP) repository on a 0.5° x 0.5° resolution (Lange and Büchner (2021)). From within the ISIMIP ensemble, data of the MRI-ESM2.0 Earth system model was chosen, since it features a medium climate sensitivity and thus gives a temperature response that best

correspond to the ensemble mean. We also use corresponding yearly atmospheric $CO_2$ concentration data from ISIMIP. We use all scenarios available from the ISIMIP: SSP1-RCP2.6, SSP3-RCP7.0, and SSP5-RCP8.5, as well as a counterfactual Control scenario that was created with constant, preindustrial $CO_2$ concentrations of 285 ppm, but recreates non-$CO_2$ forcing and interannual variability (Fig. 1a and Table 1) Over the study region, both mean temperature and mean precipitation increase in all climate warming scenarios (Fig. A1). The climate response for the SSP3-RCP7.0, and SSP5-RCP8.5 is very similar, despite

their differences in $CO_2$ levels. Temperature increases are most pronounced in the winter months (Fig. B1a). In the control scenario, average temperatures remain below 0 °C from October to April, in the highest warming scenario SSP5-RCP8.5 from November until March. Precipitation changes are stronger in the summer but also show higher interannual variability (Fig. B7b). We use soil data from the Harmonized World Soil Database, aggregated to the resolution of the climate data (FAO and IIASA (2023)). The model assumes yearly nitrogen deposition of 750 gha$^{-1}$ (Tian et al. (2018)).

## 2.3  Modeling Protocol

We combined each climate warming scenarios with a range of disturbance regimes. Equally, we combined all disturbance scenarios with a counterfactual control climate simulation to account for interannual and decadal variability in climate. We thus simulated 16 scenarios in total (see Table 1 and Fig. 1b for an overview). To reduce dimensionality, for the purpose of this study we describe disturbance regimes based on disturbance probability while keeping intensity and size constant. We chose disturbance probabilities to span from the low end of what is historically observed (return intervals of $\sim 300$ years, see e.g. Burrell et al. (2022) and Rogers et al. (2013) to the high end of what is historically observed and projected for the future (return intervals up to 10 years, see e.g. Buma et al. (2022); Burrell et al. (2022) or Turner et al. (2019).

We simulate all gridcells within the boreal forest (taiga) biome as defined by the WWF classification of terrestrial ecoregions of the world (Olson et al. (2001)) predominantly covered by needleleaf evergreen trees after the spinup period. We considered needleleaf evergreen trees to be dominant if that PFT constituted the maximum share of above-ground carbon (AGC) or fractional plant cover (FPC) in that grid cell.

The simulation setup is given in Fig. 1a. First, we spun the model up for 1000 years, recycling the pre-industrial climate of 1850 - 1879. During spinup, we prescribed a disturbance probability $p_D$ of $0.00\overline{3}$ (Return interval (RI) of 300 years). We chose this low end of observed disturbance, as it allows us to create a largely undisturbed but ecologically realistic setup needed to separate disturbance from climate effects during the simulation period. In the climate warming scenarios, we next simulated historical warming until the year 2015 while keeping $p_D$ constant, after which we saved the simulation state. We then restarted the simulation from the state, running the different model configurations of climate-disturbance combinations until the year 2100 (experimental phase). In 2100, we switched the disturbance probability back to baseline (the same as during spinup) and ran the model until 2500 to observe recovery (spindown phase). For this, we created time series of constant end-of-century temperature, precipitation, and radiations, randomly sampling the data of the years 2095 – 2100 to account for interannual variability. We used CO2 from the year 2100 for the whole period.

In the case of the control simulations, we followed the same protocol but used counterfactual climate based on pre-industrial $CO_2$ concentration throughout.

## 2.4  Data analysis

### 2.4.1  Vegetation composition

We analyze vegetation composition in terms of fractional plant over (FPC) and aboveground carbon (AGC). FPC describes the fraction of soil covered by a specific PFT. If FPC is smaller than one, vegetation does not cover the soil completely, and the bare soil fraction is calculated as $1 - FPC_V$. FPC can be larger than 1 in the case of dense, multi-layered vegetation. We chose FPC as our main variable of interest as it most directly influences the later calculation of the land-surface properties albedo and evapotranspiration. Within this study, we use the term FPC when soil fraction is included (so this can be larger than one). We use the fraction of vegetation cover $\chi^V$ to express which percentage of vegetation FPC (excluding bare soil) consists of which PFT:

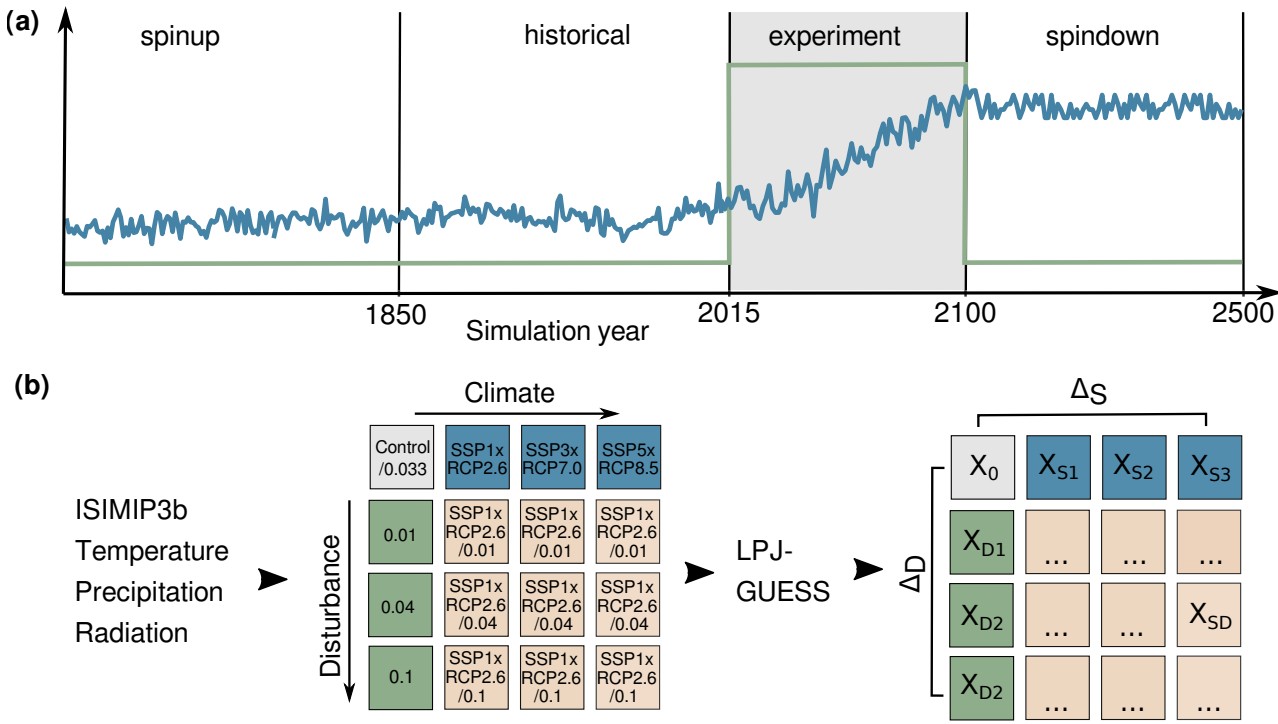

**Figure 1.** Simulation setpup. (a) The elements of a simulation run: spinup, historic, experimental and spindown phase. Blue line indicates a example climate scenario represented by mean annual temperature, green line a disturbance regime. Both trajectories are illustrative and not true to scale (see Figure A1 for timeseries of climate variables). (b) Overview over the factorial experiments performed and the methodology for driver attribution.

**Table 1.** Experimental setup. In the following, we refer to different climates as *scenarios*, disturbances as *regime* or probability, and a climate-disturbance combination as a (model) *configuration*. See also Fig. 1. $\overline{\Delta T}$ is relative to the Control scenario

| Climate scenarios | | | | | |
|---|---|---|---|---|---|
| Scenario | Description | $\overline{\Delta T}$ in K (study region) | | | |
| CO$_2$ in ppm | | | | | |
| | | 2040-2070 | 2070-2100 | 2100 - 2130 | in 2100 |
| Control | Counterfactual control | – | – | – | 285 |
| SSP1-RCP2.6 | Low warming, compatible with global 2° C target | 3.1 | 2.3 | 1.53 | 446 |
| SSP3 - RCP70 | High warming | 2.4 | 3.6 | 4.8 | 867 |
| SSP5-RCP8.5 | Very high warming | 4.3 | 5.9 | 6.2 | 1135 |
| **Disturbance regimes** | | | | | |
| Probability: | 0.00$\overline{3}$ (*baseline*) | 0.01 | | 0.04 | 0.1 |
| RI: | 300 | 100 | | 25 | 10 |

$$\text{FPC}_V = \sum_i^I \text{FPC}_i \quad \forall \quad \text{PFTs } i$$

$$\chi_i^V = \frac{\text{FPC}_i}{\text{FPC}_V}. \tag{1}$$

For clarity of analysis, we combine all shrub and non-woody PFTs into one non-tree vegetation type. Further, we combine the PFTs BNE (*Boreal needleleaf evergreen*) and BINE (*Shade-intolerant boreal needleleaf evergreen*) to represent all boreal needleleaf evergreen (NL) trees. In some instances, we additionally combine the PFTs IBS (*Shade-intolerant (pioneering) broadleaf summergreen*) and TeBS (*Temperate broadleaf summergreen*) into one broadleaf summergreen (BL) category.

     To account for interannual variability, the end-of-century state is represented by the mean and standard deviation over the

years 2085 to 2100. For all other analyses, we smooth data with a 30-year window.

     We validate our simulated historical aboveground carbon by comparing it to LiDAR-based estimates of aboveground biomass across the study region, taken from NASA's NACP campaign (Neigh et al. (2013, 2015); Margolis et al. (2015)). The dataset uses measurements of the years 2005 and 2006 to obtain a point estimate per grid cell. We regridded the data to the LPJ-GUESS resolution of 0.5° and converted it from biomass to aboveground carbon using a conversion factor of 0.5 (Pugh et al. (2024);

Sandström et al. (2007)). We used aboveground biomass/carbon for validation, as carbon cycle indicators are the best-suited diagnostic variables to assess the model performance of LPJ-GUESS.

### 2.4.2   Biophysical land-surface properties

We calculate monthly albedo $\Lambda$ of a grid cell as the sum of the characteristic seasonal albedo values of different vegetation types $\Lambda_i$, multiplied by their respective FPC (adapted from Gregor et al. (2022) and Miller and Smith (2012)):

$$\Lambda = \sum_i^I (\chi_S \Lambda_{i,S} + \chi_0 \Lambda_{i,0}) \text{FPC}_i + (\chi_S \Lambda_{Soil,S} + \chi_0 \Lambda_{Soil,0}) \chi_{Soil} \tag{2}$$

where $\chi_S$ indicates snow-covered fraction, $\chi_0$ snow-free fraction and the soil fraction $\chi_{Soil}$

$$\chi_{Soil} = \max(0, 1 - \chi^V) \tag{3}$$

where $I$ is the number of PFTs.

     Characteristic albedo values are taken from Boisier et al. (2013) (see also Table 2). We classify BNS trees as broadleaf

summergreen as previous studies show that they are closest in specific albedo (e.g. Hollinger et al. (2010)).

     LPJ-GUESS outputs the average snow depth $h_S$ in cm. We calculate the snow cover fraction $\chi_S$ from this as

$$\chi_S = \frac{h_S}{0.1 + h_S} \tag{4}$$

following Wang and Zeng (2010).

**Table 2.** Characteristic albedo values $\Lambda_i$ for different land cover types adapted from Boisier et al. (2013).

| Land cover | Summer (Mar. - Sep.) | | Winter (Oct. - Feb.) | |
|---|---|---|---|---|
| | snow-covered | snow-free | snow-covered | snow-free |
| Evergreen forest | 0.205 | 0.104 | 0.205 | 0.094 |
| Summergreen forest | 0.244 | 0.117 | 0.244 | 0.153 |
| Shrubs & Grasses | 0.568 | 0.161 | 0.568 | 0.176 |
| Soil | 0.535 | 0.246 | 0.535 | 0.205 |

The model outputs monthly transpiration, soil evaporation, and leaf interception. Total evapotranspiration is calculated as the sum of the three.

Both albedo and evapotranspiration show high interannual variability. Therefore, we performed a two-sided Wilcoxon signed-rank test on a per-gridcell basis to asses if albedo and evapotranspiration significantly differed from the configuration Control/0.003 over the years 2070 - 2100 ($p < 0.01$).

### 2.4.3 Attribution of drivers

To attribute impacts to drivers in combined climate-disturbance configurations, we assume the observed total effect $\Delta_{SD}$ to be the combination of a climate effect $\Delta_S$, a disturbance effect $\Delta_D$, and an effect $\Delta_X$ representing interactions and other non-linearities (following Verbruggen et al. (2024)).

$$\Delta_{SD} = \Delta_S + \Delta_D + \Delta_X \tag{5}$$

We define an effect $\Delta_i$ as

$$\Delta_i = x_i - x_0 \tag{6}$$

where $x_0$ is the control model state, and $x_i$ is the model state of a configuration $i$. From our factorial experiments, we can calculate $\Delta_S$, $\Delta_D$, and $\Delta_{SD}$ directly (see Figure 1b) and $\Delta_X$ from there as

$$\Delta_X = x_0 + x_{SD} - x_S - x_D \tag{7}$$

### 2.4.4 Tools

LPJ-GUESS simulations were performed on the CoolMuc2 Linux Cluster of the Leibniz Supercomputing Center, Munich. All data analyses are executed in the R programming language (R Core Team (2022)) in RStudio Version 2022.12.0 using the packages `tidyverse` 1.3.2. (Wickham et al. (2019)), `furrr` 0.3.1 (Vaughan and Dancho (2022)), `sf` 1.0.9 (Pebesma and Bivand (2023); Pebesma (2018)), `terra` 1.7.3 (Hijmans (2023)), and `rnaturalearth` 0.3.2. (Massicotte and South

(2023)). Plots are created with `ggplot2` (Wickham (2016)) and `cowplot` 1.1.1 (Wilke (2020)) using the Crameri color
scales (Crameri et al. (2020)) as implemented by Pedersen and Crameri (2022).

## 3  Results

### 3.1  Vegetation composition

Following the spin-up phase, the study region's vegetation is predominantly comprised of needleleaf evergreen trees, account-
ing for 80 % of the AGC and 59 % of vegetation cover (excluding bare soil fraction, so $\chi^V$, see also Table B1). Broadleaf
summergreen trees (10 % of AGC and 5 % of vegetation cover) and non-tree vegetation (6 % of AGC and 34 % of vegetation
cover) are relevant subdominant populations in our simulations. The range of total AGC compares well to satellite-derived
data (Fig. B2); however, observed values are lower and feature a pronounced peak around 3 $kg\,m^{-2}$, while the distribution of
modeled data is broader. LPJ-GUESS tends to overestimate aboveground carbon in Western Canada, Scandinavia, and Western
Russia and underestimate in Siberia (Fig. B3).

Keeping the disturbances constant at baseline, warming reduces the bare soil fraction from 42 % in the control scenario to
27 % in the highest warming scenario SSP5-RCP8.5 by 2100 (first bar of each block in Fig. 2A). Consequently, the fraction of
vegetation cover increases. Vegetation composition changes moderately across climate scenarios The relative contribution of
non-tree vegetation to vegetation cover decreases, while that of both needleleaved and broadleaf summergreen trees increases.
In the strongest warming scenario SSP5-RCP8.5, vegetation by the end of the century is composed of 69 % needleleaved trees,
21 % broadleaved trees, and 9 % non-tree vegetation (compared to 57 %, 8 % and 34 % in control simulations). However, in
terms of dominant vegetation cover, we see little change across climate scenarios. The majority of grid cells remains domi-
nated by needleleaf evergreen trees, while small areas of the Southern ecotone transition to dominance of either pioneering or
temperate broadleaf summergreen species (first panel Fig. 2c and first column of Fig. B4).

Overall AGC increases with warming from 3.6 $kg\,m^{-2}$ to 7.5 $kg\,m^{-2}$ (first bar of each block in Fig. 2b). Broadleaf summer-
green trees gain above average (from 19 % to 32 % in the highest warming scenario SSP5-RCP8.5) on the expense of non-tree
vegetation and needleleaf evergreen who decrease their AGC shares from 4 % to 1 % and 75 % to 67% respectively.

In contrast, keeping climate constant but increasing disturbance probability barely affects the bare soil fraction but strongly
impacts vegetation composition (first block of Fig. 2a). Disturbances strongly reduce the share of needleleaf evergreen trees
until they arrive at 5 % of vegetation cover (and 3 % of total FPC when including soil) for the highest disturbance probability
of $p_D = 0.1$ in the year 2100. Non-tree vegetation make up 91 % of all vegetation cover by the end of the century in this
configuration, and broadleaf summergreen trees 4 %. Consequently, the vast majority of grid cells are dominated by non-tree
vegetation by the end of the century in this disturbance regime for all climate scenarios (center panel of Fig. 2c, and right
column in Fig. B4).

AGC is strongly reduced by disturbances from 3.6 $kg\,m^{-2}$ to 0.5 $kg\,m^{-2}$ for the highest disturbance probability (first block
of Fig. 2b). This happens mainly at the expense of trees, while non-tree vegetation gain carbon in both relative and absolute
terms.

We see different dynamics in the case of the combined climate-disturbance configurations. In a warmer climate, an increase in disturbance leads to a further reduction of bare soil, for example, reaching 24 % in the high-warming/high-disturbance configuration SSP5-RCP8.4/0.04, compared to 27 % through high warming alone (Fig. 2a). For the highest disturbance probability of 0.1, soil fraction increases again to 30 %. In terms of vegetation composition (see also Figure B4), the disturbance-induced replacement of needleleaf evergreen trees with non-tree vegetation remains the dominant pattern. However, we see the opposing warming-induced increase of broadleaf summergreen trees, which is further exacerbated by disturbance. This increase is non-linear, reaching its peak for the second-highest disturbance regime $p_D = 0.04$, where broadleaf summergreen trees make up 32% of vegetation cover and 24 % of total FPC (compared to 21 % and 15 % respectively through warming alone). For the highest disturbance scenario of 0.1, the share of broadleaf summergreen trees is again comparable to the baseline disturbance. We see a similar effect in terms of AGC, where broadleaf summergreen species, for example, make up 54 % of AGC in the combined SSP5-RCP8.5/0.04 configuration by 2100, compared to 32 % through warming alone.

The higher absolute and relative share of pioneering broadleaves translates to a shift towards broadleaf summergreen dominance in distinct, mostly southern regions of the study domain (right panel in Fig. 2c). The number of such shifts increases with disturbance and climate (Fig. B4). The majority of remaining grid cells remain at needleleaf evergreen dominance for a $p_D$ of 0.01 and transition to predominantly non-tree vegetation for a $p_D$ of 0.04.

To further disentangle the role of the different drivers in the combined configuration, we next perform the factorial attribution and investigate the relative contribution of drivers over time (Figure 3). Here, we focus on the configuration SSP5-RCP8.5/0.04, which showed the strongest interaction effect between climate and disturbance. The configurations SSP5-RCP8.5/0.1 and SSP1-RCP2.6/0.1 are given in the Appendix (Figures B5 and B6).

In the climate-only configuration for the SSP5-RCP8.5/0.04 scenario, continues to do so until the end of the experimental period (blue lines in Figure 3a). At the end of the experimental period, $\chi^V$ declines again and stabilized after 200 years at a $\Delta_S$ of 0.1. Needleleaf tree cover $\chi_{NL}$ equally increases and reaches peak $\Delta_S$ of 0.2 mid-experimental phase, before declining again (3b). $\chi_{BL}$ again reaches pre-industrial levels 100 years into the spindown period and at the end of the simulation $\Delta_S$ is -0.1. By contrast, broadleaf tree cover $\chi_{BL}$ shows a steady increase, by 0.13 at the end of the experimental phase and by 0.37 by the end of the spindown phase (3d). $\chi_{non-tree}$ decreases from the start of the experimental period (3f). By the middle of the experimental phase, it has declined by -0.15 and stays around this value for the rest of the simulation. The response for SSPRCP2.6 is overall weaker, but show similar patterns (Figure B5)

Disturbances do not have an effect on $\chi^V$, as also visible from Figure ??a. $\chi_{BL}$ is strongly reduced by disturbance, in this configuration by -0.24 at the end of the experimental period. After disturbance pressure is lifted, it takes 200 years for $\chi_{BL}$ to recover. $\chi_{BL}$ is not affected by disturbances, except a small increase in the century after disturbance pressure is lifted. $\chi_{non-tree}$ strongly increases, by 0.26 by the end of the experimental phase. Once disturbance pressure is lifted, this reverts first fast and then slowly, reaching pre-experimental levels after 250 years. Dynamics for the higher disturbance probability $p_D$ of 0.1 look similar, but here total vegetation cover is reduced during the experimental phase, driven by a stronger decimation of evergreen tree cover (Figure B5 and B6).

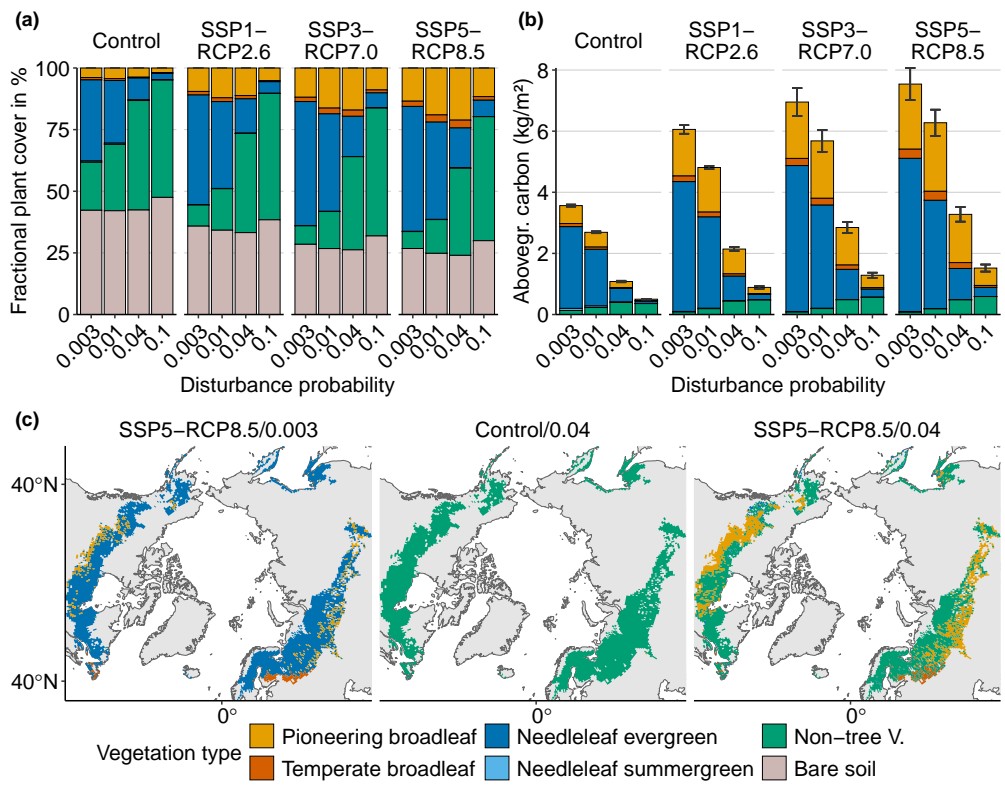

**Figure 2.** End-of-century vegetation composition across the study domain. **(a)**: Mean FPC by PFTs for all scenarios across the study domain. Mean vegetation FPC is always smaller than 1, and bare soil fraction is calculated as $1 - FPC_V$. Therefore, absolute FPC equals relative FPC **(b)**: Above ground carbon (AGC) per PFT and model configuration. Bars indicate mean over years 2700 - 2100; error bars standard deviation. **(c)**: Spatial patterns of end-of-century dominant vegetation (defined as the largest share of FPC per grid cell) exemplary for a high-warming/control-disturbance (left), control-climate/high-disturbance (middle) and high-warming/high-disturbance configuration (right).

In the combined configuration SSP5-RCP8.5/0.04, increases in $\chi^V$ exceed those of the pure climate forcing. Since disturbances do not have an impact for a $p_D$ of 0.04, there is a small interaction effect of 0.03 by the end of the experimental phase and by 0.07 at peak levels 70 years after the experimental phase ends. However, the decrease in $\chi_{BL}$ is stronger than what would have been expected from the net of climate-driven increase and disturbance-driven decreases, leaving an interaction of 0.09 by the end of the experimental phase. In contrast, the combined increase in $\chi_{BL}$ is twice the climate-driven effect by the end of the experimental phase and more than three times $\Delta_S$ at peak levels 30 years later, creating an interaction effect of 0.15 and 0.26 respectively. A notable difference here emerges in the case of the SSP1-RCP2.6 scenario, where the interaction effect is much smaller. The picture is less clear in the case of $\chi_{non\text{-}tree}$. $\Delta_{SD}$ followes $\Delta_D$ very closely for the first decades of the experimental period before leveling off and declining sharply after the end of the disturbance period. Therefore, the disturbance effect is positive during the experimental period, 0 at its end and negative (maximum $\Delta_x$ of -0.13) during the spindown phase. The differences between scenarios are again small in this case.

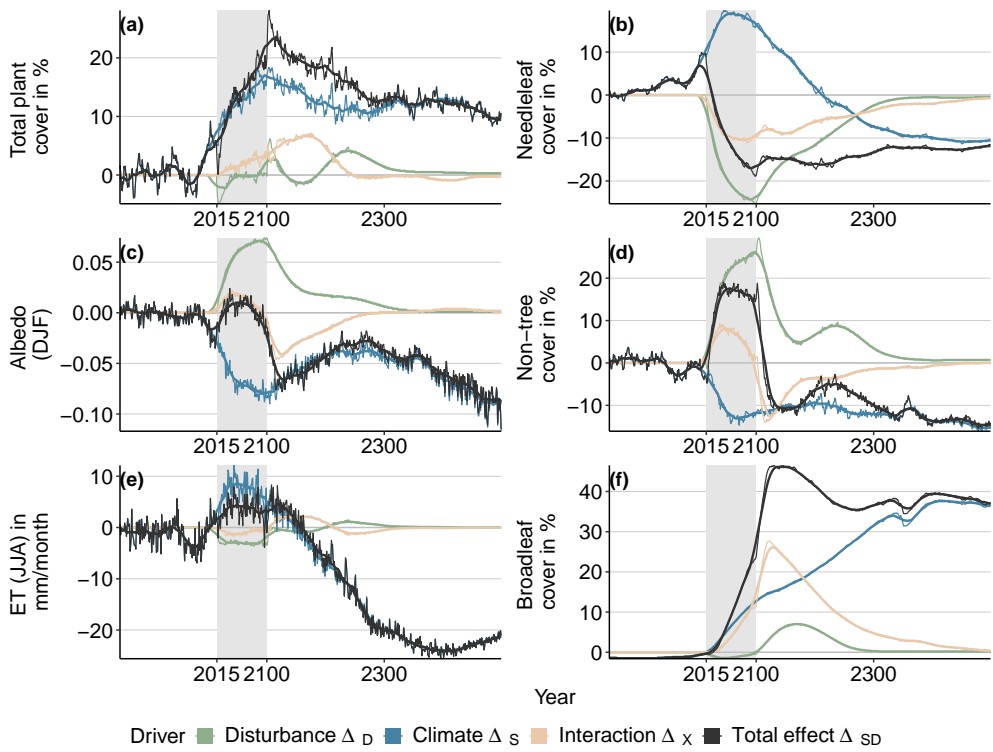

**Figure 3.** Total effects relative to control conditions and their attribution to different factors for vegetation composition, albedo, and evapotranspiration for the model configuration SSP5-RCP8.5/0.04. Grey boxes indicate the experimental period during which the disturbance regime is changed.

By the end of the simulation (year 2500), all disturbance-related effects have disappeared ($\Delta_{SD} = \Delta_S, \Delta_D = 0, \Delta_X = 0$), and vegetation is in equilibrium again ($\Delta_{SD} = const$) in all cases, with little differences between scenarios.

## 3.2 Surface properties

### 3.2.1 Albedo

Albedo exhibits pronounced seasonality. Under control conditions, it reaches peak values of $0.36 \pm 0.005$ in March and minimum values of $0.18 \pm 0.01$ in October (Fig. 4a). Simulated winter albedo agrees well with recent observations, while summer albedo in our study is on the high end of observations (see Table 3). Warming alone decreases albedo, especially in winter, where maximum values at the end of the century are reduced to $0.33 \pm 0.003$ in the low warming scenario SSP1-RCP2.6 and by $0.25 \pm 0.005$ in the highest warming scenarios SSP5-RCP8.5 (solid light and dark blue in Fig. 4a). The seasonal amplitude in albedo decreases with warming, from $0.18 \pm 0.006$ in the control climate to $0.16 \pm 0.002$ in a low-end warming scenario (SSP1-RCP2.6) and $0.14 \pm 0.001$ in a high-end warming scenario (SSP5-RCP8.5). Albedo reductions are visible

**Table 3.** Comparison of simulated monthly values for albedo and evapotranspiration under control/ historical climate to observed values.

| Albedo | DJF | JJA | |
|---|---|---|---|
| **Own simulations** | **0.33** | **0.18** | |
| Kim et al. (2024) | - | 0.21 | *North America, ABoVE domain, pre-fire* |
| Potter et al. (2020) | 0.31 | 0.11 | *North America, 60 years after disturbance* |
| Rogers et al. (2013) | 0.33 | 0.08 | *North America, mature forest stands* |
| Kiljunen (2006) | 0.35 | 0.12 | *Finnland, needleaf evergreen stands* |
| ET in mmmonth$^{-1}$ | DJF | JJA | |
| **Own simulations** | **0.3** | **69.0** | |
| Wang et al. (2021) | 3.7 | 83.4 | *circumboreal, 1982–2015 mean, modelled* |
| Ju et al. (2010) | 6.8 | 73.9 | *Canada, various sites, Eddy covariance* |

throughout the study region. There are, however, spatial variations in magnitude, with distinct patches of strong anomalies in Eastern Canada and Eurasia, while other areas, especially in Western Canada, Alaska, and Southern Russia, show very little change (left panel in Fig. 4c).

An increase in disturbance probability alone has the opposite effect. Increasing $p_D$ from 0.003 to 0.04 while keeping climate constant increases winter albedo to $0.40 \pm 0.005$ in winter and to 0.21 in summer (dashed pink line in Fig. 4a and pink bars in Fig. 4b). The seasonal amplitude increases to 0.23. The magnitude of change is more uniform throughout the study region (center panel in Fig. 4c).

In the combined scenarios, the pattern of increasing albedo with disturbance probability and decreasing albedo with warming is preserved. However, the net effect differs between scenarios. For the moderate increase in disturbance probability $p_D$ = 0.01, the climate effect prevails, resulting in a net decrease in albedo (second group in Fig. 4b). For the highest disturbance probability $p_D$ = 0.1, the disturbance effect is stronger, leading to a net increase in albedo compared to baseline disturbance (right group in Fig. 4b). In the middle case of $p_D$ = 0.04, we observe a net increase for the SSP1-RCP2.6 scenario and a slight net decrease for the SSP5-RCP8.5 scenario. In the winter months SSP5-RCP8.5/0.04 is almost at par with the control/baseline configuration (dashed dark blue line in Fig. 4a). This change is not uniform across the domain (right panel in Fig. 4c). The climate-disturbance configuration SSP5-RCP8.5/0.04, for example, shows an albedo increase in distinct regions and decreases in others, resulting in the small net change visible in Fig. 4a and a. While the pure climate- and disturbance effect results in significant changes in the majority of the study region, the combined effect can not be separated from interannual variability in this particular configuration.

When investigating the different albedo drivers over time, again for the example configuration SSP5-RCP8.4/0.04, shows that the climate and disturbance effects constantly increase over the scenario and maintain comparable orders of magnitude but in different directions (Figure 3c). At the end of the experimental period, the climate effect is -0.079, and the disturbance effect is 0.068. The interaction effect is small at this point (-0.01). The total albedo effect is, therefore, negligible for most of the scenario ($< 0.01$), only declining in the last decade of the experimental period to reach -0.023 in 2100. These trends are

reversed after the end of the scenario. $\Delta_{SD}$ converges with $\Delta_S$ while $\Delta_D$ declines but only approaches 0 after the year 2300. Consequently, we see a counteracting interaction effect until this point as well. The final net albedo effect in simulation year 2500 is -0.088. In both configurations of the high disturbance probability 0.1, the net albedo increased by up to 0.07, due to stronger disturbance-mediated increases but also interactions effects (Figure B5 and B6).

### 3.2.2 Evapotranspiration

Like albedo, evapotranspiration shows high seasonality (Fig. 5a). For all configurations, evapotranspiration is low ( < 1 mm/-month) in winter (DJF) and reaches highest levels in July. Peak evapotranspiration reaches 69 $\pm$ 2 mm/month in the control/baseline scenario. Additionally, evapotranspiration shows the strongest interannual variation. Our simulated values are slightly lower than observations but capture the seasonal amplitude well (see Table 3).

Warming alone increases evapotranspiration. The strongest effect is seen in spring, where ET increases by about 4.9 $\pm$ 2.3 mm/month for the low-end warming scenario SSP1-RCP2.6 and 10.2 $\pm$ 3 mm/month in the highest warming scenarios SSP5-RCP8.5 (solid light and dark blue in Fig. 5a and left group in Fig. 5). Notably, maximum evapotranspiration does not differ between the low-end and the high-end warming scenario (76.4 $\pm$ 2 mm/month and 76.5 $\pm$ 3 mm/month respectively). Climate-induced change in evapotranspiration is seen across the study domain, but the magnitude varies (first panel of Fig. 5c). The strongest decrease is seen in Eastern Canada and South-Eastern Russia.

Disturbance alone decreases evapotranspiration (dashed pink line in Fig. 5a and pink bars in Fig. 5), for the most intense disturbance regime by -3.47 $\pm$ 0.25 mm/month. This decrease is concentrated in distinct areas, while most of the study regions shows no significant change (middle panel of Fig. 5c).

For the majority of combined climate-disturbance configurations, the net effect is an increase in evapotranspiration (Fig. 5b). The exception is SSP1-RCP2.6/0.1 (low-end warming scenario), where evapotranspiration remains unchanged from baseline, as the climate and disturbance effects offset each other. The spatial analysis shows that, again, evapotranspiration increases in most areas (right panel of Fig. 5 for SSP5-RCP8.5/0.04). However, there are distinct areas that show no significant change. Maximum evapotranspiration is 72.3 $\pm$ 2.8 mm/month and 72.7 $\pm$ 2.6 mm/month for the SSP1-RCP2.6 and SSP5-RCP8.5 scenarios respectively and for a $p_D$ of 0.04 (dashed blue lines in Fig. 5a).

Drivers of evapotranspiration effects over time show similar patterns to albedo but in reversed directions (Figure 3e). Again, for the example configuration SSP5-RCP8.5/0.04, climate increases evapotranspiration with peak levels of +8.5 mm/month around the year 2050, while disturbances reduce evapotranspiration. Contrary to albedo, $\Delta_D$ is smaller in magnitude than $\Delta_S$, and the net effect over the experimental period is therefore positive, reaching 3.1 mm/month at the end of the scenario. There is no interaction effect. The disturbance effect declines immediately after the end of the scenario. $\Delta_S$ and $\Delta_{SD}$ reverse and become negative around 50 years after the experimental period. The final effect at the end of the simulation period is -24 mm/month. Similarly to albedo, the disturbance-induced effect is stronger for a $p_D$ of 0.1, but convergence happens equally fast.

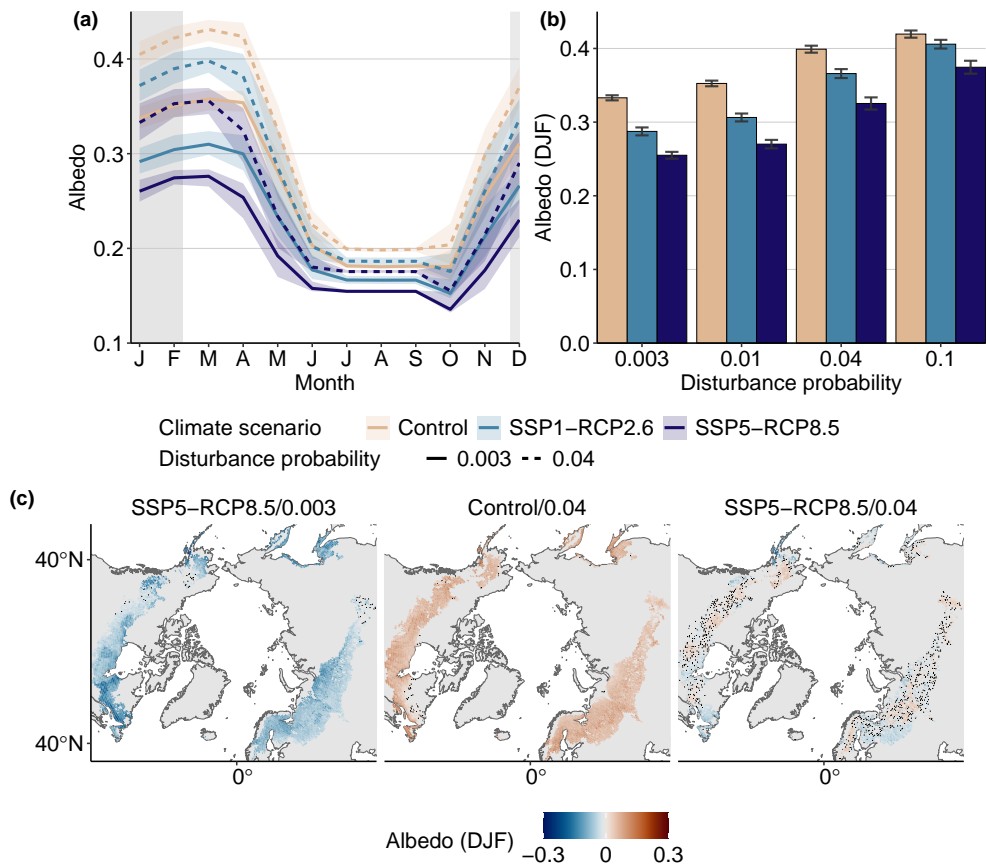

**Figure 4.** End of century albedo across the study domain. **(a)**: Seasonal albedo for selected configurations. Pink lines indicate control climate and blue lines show low-end (light blue) and high-end (dark blue) warming. Solid lines indicate baseline disturbance regimes, dashed lines a $p_D$ of 0.04. Thick lines show mean over 2070 - 2100, ribbons range over individual years. **(b)**: Winter (DJF) albedo for a range of configurations. Bars indicate mean over the years 2070 - 2100, error bars $\pm$ one standard deviation. **(c)**: Spatial patterns of albedo anomaly (relative to control/baseline) for a high-warming/control-disturbance (left), control-climate/high-disturbance (middle) and high-warming/high-disturbance configuration (right). Stiples indicate areas where albedo does not significantly differ from configuration Control/0.003 ($p < 0.01$).

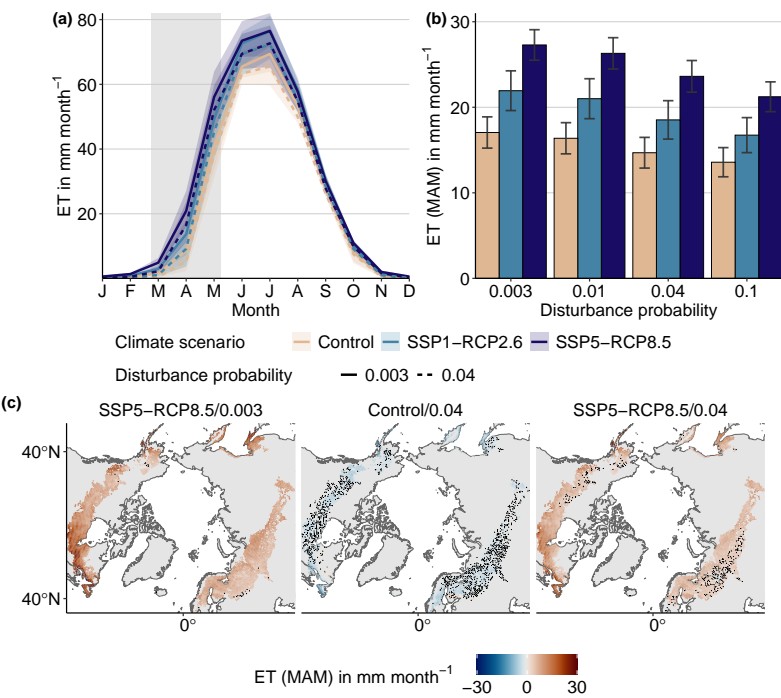

**Figure 5.** End of century evapotranspiration (ET) across the study domain. **(a)**: Seasonal ET for selected configurations. Pink lines indicate control climate, blue lines show low-end (light blue) and high-end (dark blue) warming. Solid lines indicate baseline disturbance regimes, dashed lines a $p_D$ of 0.04. Thick lines show mean over 2070 - 2100, ribbons range over individual years. **(b)**: Spring (MAM) ET for a range of configurations. Bars indicate mean over the years 2070 - 2100, error bars $\pm$ one standard deviation. **(c)**: Spatial patterns of ET anomaly (relative to control/baseline) for a high-warming/control-disturbance (left), control-climate/high-disturbance (middle) and high-warming/high-disturbance configuration (right). Stipples indicate areas where evapotranspiration does not significantly differ from configuration Control/0.003 ($p < 0.01$).

## 4 Discussion

### 4.1 Vegetation composition

Historical species distributions produced by our model are in line with observational data and previous modeling studies performed with LPJ-GUESS (see e.g., discussion of Zhang et al. (2013) or Wolf et al. (2008)), and the range of total AGC corresponds to observations (Fig. B2). The higher AGC in our simulations is respected as we simulate potential natural vegetation and do not consider impacts of land use change or harvest, which is especially important in Scandinavia and Canada (Pugh et al. (2019b); Potapov et al. (2017); Curtis et al. (2018)). Additionally, a disturbance probability of 300 years employed during

spinup is low compared to the observed frequency in some areas. For example, in Western Canada and Alaska, fire return intervals of 100 years are reported, explaining that our model set-up leads to an accumulation of aboveground carbon in these

regions, compared to observations. The chosen setup is thus a trade-off between capturing historical conditions and our goal to obtain largely undisturbed vegetation at the start of the scenario to be able to separate disturbance from climate effects during the experimental period. Species composition under historical climate is robust in our model against a change in disturbance probability within historical return intervals that rarely exceed the 100 - 50 year range (Fig. 2, Burrell et al. (2022); Rogers et al. (2013)). Therefore, it is nevertheless reasonable to assume that our simulations represent a realistic ecological state.

In general we find that climate is the dominant driver of the increase in total vegetation cover and carbon, while a complex interplay between climate, disturbance and their interactions mediates changes on the PFT level and thus vegetation composition. Climate change induces an increase in needleleaf evergreen tree cover at the expense of non-tree vegetation, as warming favours the expansion and northward migration of trees (Boulanger and Pascual Puigdevall (2021); Gustafson et al. (2021); Rees et al. (2020); Zhang et al. (2013)) while disturbances have the opposite effect, decreasing needleaf evergreen tree cover and increasing that of non-tree vegetation in our simulations. Depending on the combination of climate and disturbance regime employed, we thus can find a net replacement of non-tree vegetation with needleleaved trees, the opposite or diverging trends in different regions. In the case of broadleaved trees, climate change favours their expansion, while disturbance does not affect their FPC share. In the absence of interaction effects, disturbance thus does not affect total vegetation cover, since the replacement of needleleaf evergreen trees with non-tree vegetation results in zero net change. Climate in turn has a net positive effect, as the expansion of both needleleaf evergreen and broadleaf summergreen trees exceeds what is being replaced in non-tree vegetation.

The interaction between climate and disturbance leads to a combined response that for needleleaf evergreen trees and non-tree vegetation is closer to the sole disturbance effect as would be expected. However, for non-tree vegetation this is reversed after the scenario end and disturbance effects quickly disappear. Both can be explained by the strong expansion of broadleaf summergreen trees in the combined scenarios. Broadleaf trees substitute both needleleaf evergreen tree and non-tree vegetation after disturbance, preventing their respective climate- and disturbance-driven expansion. Given the assumption that future disturbance rates will significantly surpass historical levels, a decline in needleleaf evergreen tree cover is likely. This is also in line with trends observed over the last decades, e.g. by Wang et al. (2020), who observed a decrease in needleleaf evergreen tree cover and an increase in broadleaf summergreen trees and non-tree vegetation over the years 1984–2014.

Spatial patterns of broadleaf summergreen tree dominance in our simulations correspond to observations from recent field surveys in North America that report such vegetation shifts predominantly in Alaska and Western Canada, while the Eastern Canadian Shield and Plains show higher rates of recovery and shifts between different needleleaved species (Baltzer et al. (2021), Figure 2c and B4). Our model additionally projected state shifts in Southern Russia in our simulations, from where comparable field surveys are still lacking. Pioneering broadleaves, such as Aspen or Birch species, are an integral part of succession cycles in many ecosystems of the boreal region (Pfadenhauer and Klötzli (2020)). Thus, one might anticipate their expansion at elevated disturbance levels solely based on a higher proportion of vegetation in an early-successional state in the model. However, when the disturbance rate was increased under a controlled climate, it had minimal impact on these species' absolute or relative abundance. Consequently, their rise cannot be solely attributed to this. Instead, a shift in climatic

conditions is additionally needed to render broadleaved species more competitive in post-disturbance recovery (Baltzer et al. (2021); Mekonnen et al. (2019); Wårlind et al. (2014)).

An effect that might not be expected at first glance is the resilience of total vegetation FPC to disturbance (Figure 2a), as reduction of vegetation density due to high disturbance is likely. We explain this finding in our simulations through the replacement of tree cover with non-tree vegetation. Therefore, while aboveground biomass rapidly decreases, FPC increases due to the higher relative leaf area of non-tree vegetation. This also explains the increase of vegetation cover and reduction of bare soil with increasing disturbance rate. Undisturbed vegetation composition in our simulations is remarkably resilient against climate change, which also corresponds to recent observations (Kim et al. (2024); Sulla-Menashe et al. (2018)). The same is not true of AGC, which is strongly diminished by disturbances (Figure 2b). Again, this affects mainly needleleaf evergreen trees, while broadleaf summergreen trees and non-tree vegetation are resilient to disturbance and increase their AGC share, also in line with recent field observations (Baltzer et al. (2021); Mack et al. (2021)). As our analysis focuses on above-ground processes, it does not allow for further conclusion regarding the impact on the boreal carbon balance, where belowground processes play an important role.

Overall, our results suggest that modeling results of future vegetation distributions are highly sensitive to the choice of disturbance regime. Therefore, without an accurate representation of disturbance regimes, there is the danger of overestimating the stability of future vegetation. Due to the interaction effects between climate and disturbances, this sensitivity becomes increasingly important with warming, while historical simulations are more robust.

## 4.2 Land surface properties and potential climate feedbacks

Our simulated historical albedo is high compared to observations in summer, while winter albedo, the main focus of our analysis, shows high agreement. Our results indicate that disturbance and climate have significant but opposing effects on albedo. Therefore, depending on the climate-disturbance configuration, we may see a net increase, decrease, or little net change (Figure 4).

Previous modeling studies such as Krause et al. (2019), Zhang et al. (2018) or Zhang et al. (2013) predominantly found albedo decreases of up to -0.25, depending on the climate scenario employed and specific region. This corresponds to our finding for a moderate disturbance scenario (Fig. 4b, second group). These studies did not explicitly consider the effects of disturbance, and the assumed disturbance rates are not always reported. However, it is likely that their results predominantly capture climate effects. Our results indicate that albedo decreases may be reduced or even reversed in a high-disturbance world.

Vegetation shifts are the main driver of albedo change, most importantly the relative shifts between non-tree vegetation and needleleaf evergreen tree cover. Contrary to what previous studies have postulated (Baltzer et al. (2021); Wang and Friedl (2019); Rogers et al. (2013)), the strong increase in broadleaf summergreen tree cover, which we found in our simulations especially between the years 2100 and 2200, does not translate to an albedo increase. This can by explained by the fact the broadleaf summergreen tree expansion occurs also at the expense of non-tree vegetation and bare soil, which have higher specific albedo. It should be noted that the calculated albedo values are highly dependent on the specific albedo values used (Table 2). The values we used were not derived specifically for the boreal forest, and it is possible that they underestimated

the difference in albedo between vegetation types. Indeed, some studies report albedos of 0.6 - 0.8 in early-successional and/ or broadleaf summergreen forest stands (Kim et al. (2024); Zhang et al. (2018); Rogers et al. (2013)). Such values would be unattainable with our approach. In other studies, however, albedo values are in line with our findings, e.g. mean albedo values after disturbance not exceeding 0.5 in Potter et al. (2020). All these studies report albedo values at site level, representing a mixture of different vegetation types and soil that are not directly translatable to specific albedo values. If specific albedos for the boreal forest were available, they would greatly improve the accuracy of albedo calculations in the future.

Snow cover dynamics drive the season albedo amplitude but play a minor role for albedo changes in the climate scenarios. This might seem surprising at first since one might intuitively expect reduced snow cover due to warming. In LPJ-GUESS, warming results in earlier snowmelt and later onset of the snow season over the study area, while snow cover during winter was barely affected by warming (Fig. B7a). This is expected from the climate forcing data used (Fig. B1) and in line with 21st-century projections from CMIP5 and CMIP6 as well as previous results from dynamic vegetation models (McCrystall et al. (2021); Krause et al. (2019); Krasting et al. (2013)). Uncertainties regarding future snow cover in the climate data used will, of course, in turn, influence our albedo calculations. We here also want to note that Potter et al. (2020) found snow cover an important predictor of post-fire albedo changes when combining statistical analysis of recent fire events with climate projections.

Evapotranspiration shows opposite patterns from albedo, as here climate leads to an increase, while disturbance leads to a decrease. Additionally, in contrast to albedo, the magnitude of change is larger for climate-induced increases. Therefore - within the realistic parameter space assessed in this study - there is no configuration that would achieve a net decrease in evapotranspiration. However, the net effect can be significantly reduced compared to the pure climate effect (Figure 5).

The impact of disturbances on evapotranspiration is less well-studied than that of albedo. Previous modeling studies found overall increases in evapotranspiration throughout our study domain (Krause et al. (2019); Zhang et al. (2013)), even though Krause et al. (2019) found diverging signals when comparing several DVMs. Again, from our results, we can expect those changes to be reduced when accounting for a high-disturbance future.

Climate-driven evapotranspiration change can occur due to both direct climate effects such as temperature or precipitation increase, $CO_2$-mediated changes in water use efficiency, and climate-modulated vegetation change, which is challenging to untangle. As temperature and precipitation both increase over the course of the scenario, it is likely that climate-induced changes are at least partly driven by direct climate effects. However, as climate is held constant after 2100 we can assume any changes taking place after that to be modulated by vegetation.

We expected an increase in evapotranspiration after disturbance due to a higher share of broadleaf summergreen trees and, in turn, higher Leaf Area Index (LAI), something we cannot confirm from our results. From the individual components making up evapotranspiration, it appears to us that the reduction of needleleaf evergreen tree cover reduces interception and increases runoff, especially in spring, and thus leads to less water being available for evapotranspiration (Fig. B8). Other studies investigating evapotranspiration after fire disturbance equally found reductions both in the field (Liu et al. (2018)) and in modeling results (Bond-Lamberty et al. (2009)). It should be noted that some processes governing ecophysiological control on ET are lacking in LPJ-GUESS. Yet, in global simulations LPJ-GUESS has been shown to accurately simulate ET compared to a num-

ber of hydrological datasets (Zhou et al. (2024)). However, future model development aiming for a better representation of plant hydraulic strategies may be able to further improve model performance. (Papastefanou et al. (2020, 2022); Meyer et al. (2024)).

An increase in both albedo and evapotranspiration would result in a cooling of the land surface, while a decrease would have a reversed effect. As LPJ-GUESS was not coupled to an atmospheric model, we cannot determine the actual net effect,
especially as other processes, such as cloud formation or surface roughness, can additionally play a role (Gregor et al. (2022); Swann et al. (2010)). Previous coupled studies with LPJ-GUESS report a net warming effect due to vegetation dynamics with a high degree of intra-annual and spatial variations (Zhang et al. (2018)). Our results indicate that accounting for an increase in disturbances and resulting vegetation changes are lightly to weaken that effect.

## 4.3  Legacy effects and resilience

We find that disturbance-induced effects on vegetation are long-lasting ($\sim$ several 100 of years) but ultimately transient and reversible on the centennial time scale if disturbance pressure is lifted again. Tree cover fraction takes the longest to recover, where the total effect converges again with the undisturbed trajectory after up to 400 years.

Our results confirm that the end-of-century vegetation state does not represent an equilibrium, as already shown by e.g. Pugh et al. (2018). Importantly, this is also true for scenarios where only the disturbance interval changes (green lines in
Figures 3, B5 and B6). Therefore, we would expect these scenarios to also show sustained change post-2100 before settling in a high-disturbance-intensity equilibrium. Exploring these equilibria and dynamics would be a worthwhile avenue for additional research.

It is interesting to note that once disturbances are set back to baseline probabilities the recovery curves look quite similar across disturbance intensities and convergence times scales of trajectories with different disturbance histories differ little
across disturbance intensities and climate scenarios. This indicates that disturbance legacies are dominated by the internal successional dynamics of the vegetation more than previous dynamics. Of course, our stylized scenarios are not a likely long-term scenario as return to baseline disturbance regimes or constant climate after 2100 is very unlikely. The question of reversibility is nevertheless relevant in the context of forest management and climate mitigation.

However, recovery patterns need to be interpreted cautiously, as there are a number of limitations in the implementation
of establishment and disturbance in the model: In LPJ-GUESS there is no interaction between gridcells for reasons of computational efficiency and thus there is no lateral exchange of seeds, that would control spatial migration. Rather, migration is emulated through a background establishment rate, a small rate of establishment that will always occur as soon as PFTs are within their bioclimatic limits. This may lead to a faster northward expansion of trees than if migration would be limited by seed dispersal (Zani et al. (2023)). Additionally, the production of offspring and thus establishment can occur once a plant is
productive, there is no maturation period. This means that regeneration failure - eroding of seed banks through too frequent disturbances (Turner et al. (2019); Hansen et al. (2018)) - cannot occur by model design. Here, different reproductive strategies, e.g. serotony or resprouting, would also be important to consider (Baltzer et al. (2021); Hansen et al. (2021)). Therefore, while

our results indicate a general possibility for recovery, the conditions under which this is a realistic scenario need to be explored in more details in the future.

In this context, disturbance type is also important. We here employed a standardized disturbance event to reduce dimensionality in our simulations and ensure controlled experiments. However, it is important to keep in mind that specific disturbances can have additional effects we are not considering here. For example, wildfire affects soil conditions, e.g. through the burning of the peat layer and changes in nutrient cycling (Mack et al. (2021); Mekonnen et al. (2019)). Further, our disturbance dynamics are uniform in space and time and not linked to vegetation type, as, for example, a detailed fire disturbance module would

be. In that case, we would especially expect disturbance frequency and impact to differ with vegetation type. Broadleaf summergreen trees are less susceptible to fire, and literature suggests that broadleaf summergreen tree dominance could maintain itself through fire suppression (Hansen et al. (2021); Mekonnen et al. (2019); Johnstone et al. (2016, 2010)). Wind damage and pathogen attacks are equally linked to climate conditions and species compositions (Mitchell (2013); Seidl et al. (2017)). Additionally, there are interaction effects of disturbances. For example, deadwood from windthrow provides a habitat for bark

beetles or fuel for wildfires (Seidl et al. (2017)). Taken together, all these limitations suggest a potential overestimation of the recovery ability of vegetation in the model.

    Disturbance-induced changes in land-surface properties are mainly driven by changes in non-tree vegetation cover, less subject to legacy effects and internal dynamics than tree cover and more directly controlled by climate. Therefore, both albedo and evapotranspiration recover quickly once disturbance pressure is lifted again, and legacy effects disappear after approximately

25 years. We stress that our results inform on the potential ability to recover rather than a realistic projection of likely dynamics after the 21st century. With sustained disturbance regimes, the respective changes in land-surface properties would likely persist.

## 5   Conclusions

In this study, we investigated the relative impact of climate change, intensifying disturbance regimes, and their interaction

on boreal vegetation and land surface properties. We found that in general climate drives shifts towards denser and more forested vegetation, while disturbances reduce the prevalence of trees in favor of shrubs and grasses. The interaction between climate and disturbances increases broadleaf summergreen tree cover, causing shifts from needleleaf evergreen and non-tree vegetation to broadleaf summergreen dominance. The shifts we observe are not driven by prescribed bioclimatic limits but are an emergent feature of the model, arising from a shift in competitive balance. This highlights the ability of LPJ-GUESS to

realistically capture the influence of climate change on succession and thus post-disturbance recovery dynamics.

    In our simulations, disturbances affect albedo and evapotranspiration. In both cases, the disturbance effect opposed the pure climate effect, while interactions between the two factors played a minor role. ET is more closely coupled to direct climate effects, with vegetation changes playing a subordinate role. In contrast, vegetation shifts (due to both disturbances and warming) are the main driver of observed albedo changes. Here, disturbance-induced effects have the potential to weaken or

even reverse climate-induced changes.

Disturbances caused long-lasting legacies in vegetation composition, which only regenerated on the centennial time scale. On the opposite, albedo and evapotranspiration recovered on a decadal timescale. Therefore, while our results show the ability for disturbances to severely disrupt land surface-atmosphere interactions, they also highlight the theoretical potential for regeneration. It remains an important avenue for future research to better understand how seed dispersal and maturation would affect this regeneration potential.

We find simulated future vegetation distributions highly sensitive to the choice of disturbance regime. Due to the interaction effects between warming and disturbances, this sensitivity becomes increasingly important when moving into a high-warming future, while historical simulations are more robust against the choice of disturbance regime. Without an accurate representation of disturbance, there is a risk of misjudging future vegetation composition and resulting land surface properties.

*Code and data availability.* LPJ-GUESS model code and raw model output are archived on Zenodo (https://doi.org/10.5281/zenodo.10619524). The code to reproduce the analyses and figures is found on Github (https://github.com/lucialayr/disturbanceBorealLPJ).

*Author contributions.* LSL and AR designed the study. LSL performed the model simulated and data analyses and prepared the manuscript. KG, AK and PM contributed to model development and setup. All authors contributed to the interpretation of the results and the revision of the manuscript.

*Competing interests.* At least one of the (co-)authors is a member of the editorial board of Biogeosciences. The authors declare no further conflict of interest.

*Acknowledgements.* LSL is supported by a PhD fellowship from the German Federal Ministry of Education and Research, issued via the Heinrich Boell Foundation. LSL and AK acknowledge support from the German Federal Ministry of Education and Research via the STEPSEC project, grand number 01LS2102C. KG acknowledges funding from the VELUX foundation, Project No. 1897. The authors acknowledge the use of Grammarly and ChatGPT for English grammar and language support.

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

**Table A1.** Overview over relevant PFT-related parameters. Leaf = Leaf type (needleleaf or broadleaf); Pheno = Phenology (evergreen or summergreen); Shade = Shade tolerance; $t^{E/S}_{t,c/s,min/max}$: Min/max coldest/warmest month temperature needed for establishment/survival; $GDD5^{E}_{min,i}$ = Growing season temperature sum; $L_i$ = Longevity

| PFT | Leaf | Pheno | Shade | $t^{E}_{t,c,min}$ | $t^{E}_{t,c,max}$ | $t^{E}_{t,w,min}$ | $GDD5^{E}_{min,i}$ | $t^{S}_{t,c,min}$ | $L_i$ |
|---|---|---|---|---|---|---|---|---|---|
| BNE | NL | EG | tolerant | -30 | -1 | 5 | 500 | -31 | 500 |
| BINE | NL | EG | intolerant | -30 | -1 | 5 | 500 | - 31 | 500 |
| BNS | NL | SG | intolerant | -1000 | -2 | -1000 | -1000 | 350 | 300 |
| TeNE | NL | EG | intolerant | -2 | 10 | 5 | 2000 | 2 | 300 |
| IBS | BL | SG | intolerant | -30 | 7 | -1000 | 350 | -30 | 300 |
| TeBS | BL | SG | tolerant | -13 | 6 | 5 | 1100 | -14 | 400 |
| HSE | NL | EG | intolerant | -16 | 5 | - | 300 | -32.5 | 100 |
| HSS | BL | SG | intolerant | -32.5 | 5 | - | 300 | -32.5 | 100 |
| LSE | NL | EG | tolerant | - | 5 | - | 100 | - | 25 |
| LSS | BL | SG | tolerant | - | 5 | - | 100 | - | 25 |
| EPDS | NL | EG | tolerant | 5 | - | - | - | 0 | 100 |
| SPDS | BL | SG | tolerant | 5 | - | - | - | 0 | 100 |
| GRT | - | | | - | - | - | - | - | |
| CLM | - | | | | | | 0 | | |
| C3G | BL | - | - | -1000 | -1000 | -1000 | -1000 | 1000 | - |

**Appendix A: Methods**

**Appendix B: Results**

**Table A2.** Overview over used PFTs

| PFT | Full name | Example species |
|-----|-----------|-----------------|
| | | Trees |
| BNE | Boreal needleleaf evergreen | *Picea abies* (L.) H.Karst. |
| BINE | Boreal shade-intolerant needleleaf evergreen | *Pinus sylvestris* L. |
| BNS | Boreal needleleaf summergreen | *Larix sibirica* Ledeb. |
| TeNE | Temperate needleleaf summergreen | |
| IBS | Shade-intolerant broadleaf summergreen | *Betula pubescens* Ehrh., |
| | | *Populus tremula* L. |
| TeBS | Temperate broadleaf summergreen | *Tilia cordata* Mill., |
| | | *Ulmus glabra* Huds. |
| | | Shrubs |
| HSE | High shrub evergreen | *Juniperus communis* L., |
| | | *Pinus pumila* (Parl.) Regel |
| HSS | High shrub summergreen | *Salix* spp., *Betula nana* L. |
| LSE | Low shrub evergreen | *Vaccinium vitis-idaea* L., |
| | | *Ledum palustre* L. |
| LSS | Low shrub summergreen | *Vaccinium myrtillus* L , *Salix glauca* L. |
| EPDS | Prostrate dwarf shrub evergreen | *Vaccinium oxycoccus* L. |
| SPDS | Prostrate dwarf shrub summergreen | *Salix arctica* Pall. |
| | | Non-woody PFTs |
| GRT | Graminoid and forb Tundra | Artemisia, Kobresia, Brassicaceae |
| CLM | Cushion forb, lichen and moss Tundra | Saxifragacea, Caryophyllaceae |
| C3G | Temperate C3 grasses | Gramineae |

**Table B1.** Average Vegetation composition at the end of the spinup period (spatial and yearly mean across the years 1850 - 1880). Factorial simulations are restarted from this state.

| PFT | AGC in $\mathrm{kg\,m^{-2}}$ | FPC in $\mathrm{m^2\,m^2}$ | AGC in % ($\chi_i^V$) | FPC in % ($\chi_i^V$) |
|-----|------------------------------|----------------------------|------------------------|------------------------|
| Needleleaf evergreen | $2.8 \pm 0.033$ | $0.34 \pm 0.079$ | $80 \pm 0.251$ | $59 \pm 0.997$ |
| Needleleaf summergreen | $0.13 \pm 0.013$ | $0.01 \pm 0.001$ | $4 \pm 0.273$ | $2 \pm 0.188$ |
| Pioneering broadleaf | $0.53 \pm 0.009$ | $0.03 \pm 0.001$ | $10 \pm 0.083$ | $5 \pm 0.184$ |
| Temperate broadleaf | $0.01 \pm 1\mathrm{e}\text{-}04$ | $0 \pm 1\mathrm{e}\text{-}05$ | $0 \pm 0.001$ | $0 \pm 0.002$ |
| Temperate needleleaf | $0 \pm 0$ | $0 \pm 0$ | $0 \pm 0$ | $0 \pm 0$ |
| Non-tree vegetation | $0.12 \pm 0.004$ | $0.19 \pm 0.01$ | $6 \pm 0.352$ | $34 \pm 1.27$ |

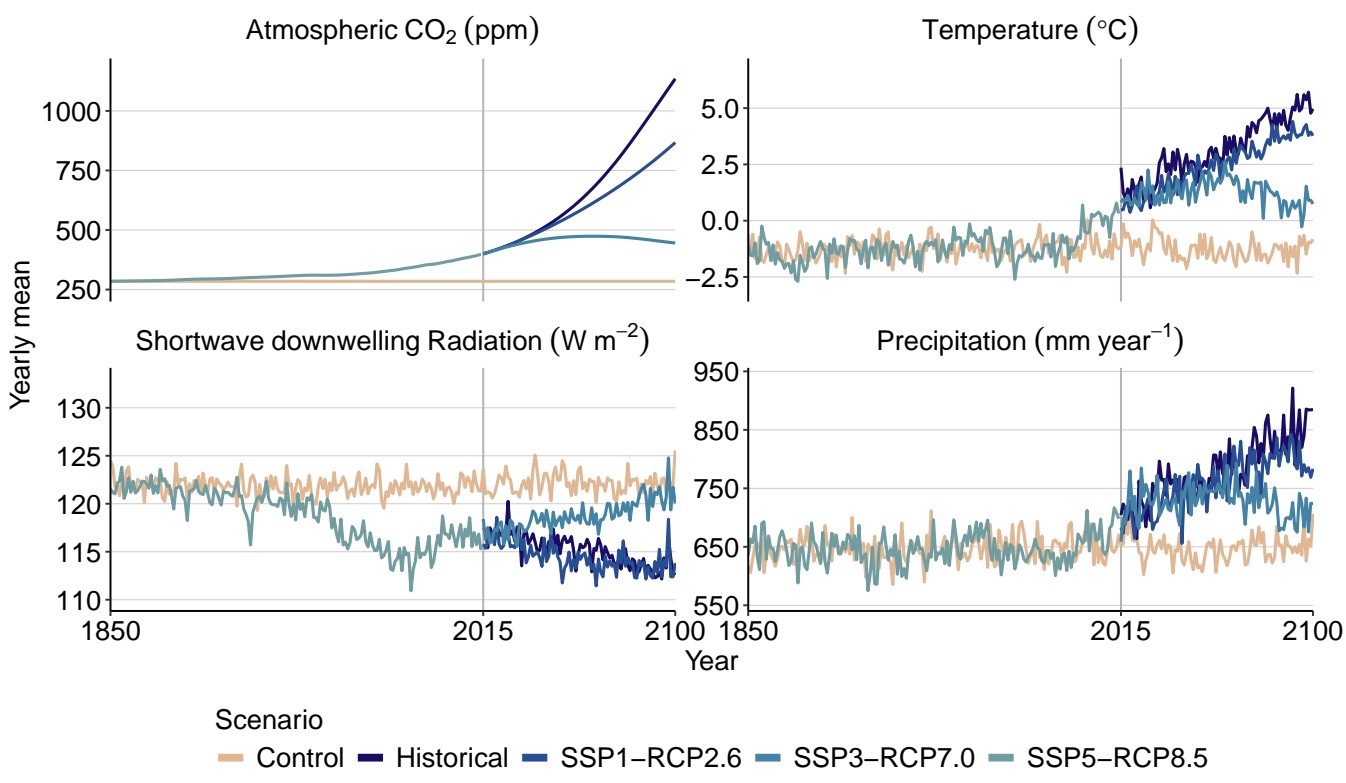

**Figure A1.** Climate data used to force the simulations. Note that model runs continue until 2500, further recycling data of the years 2095 - 2100.

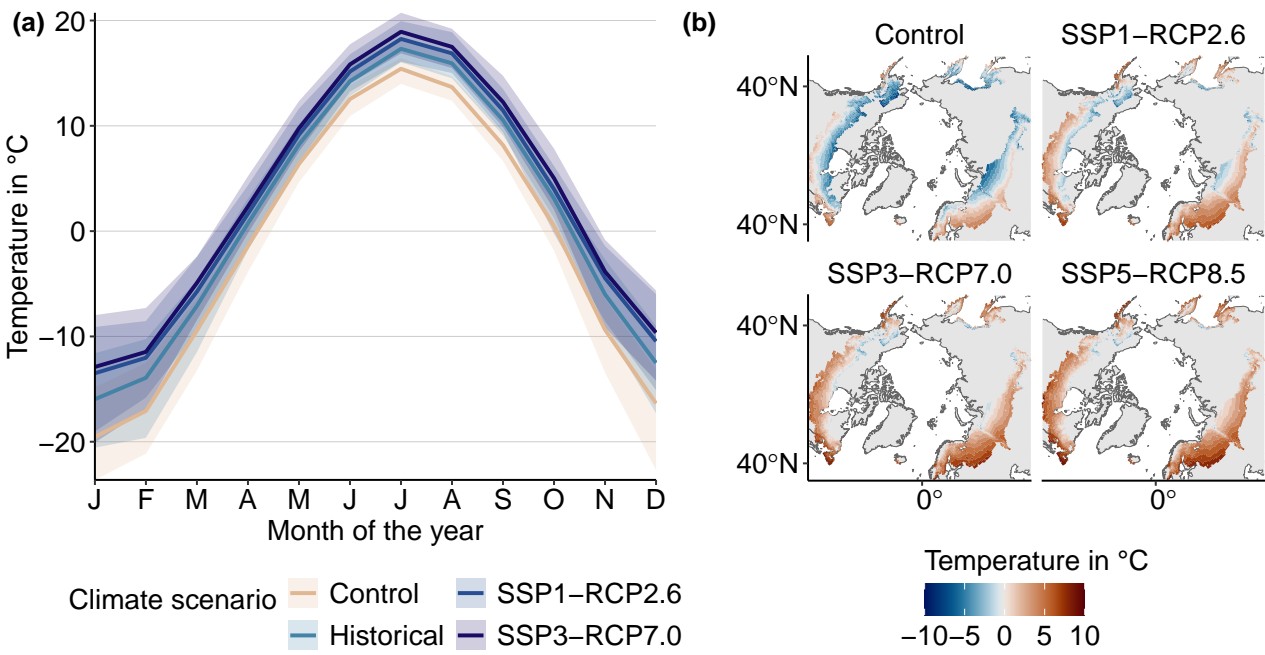

**Figure B1.** Overview over the MRI-ESM2 temperature data used to force LPJ-GUESS. (a) Seasonal curves, averaged over the years 2070 - 2100. Ribbon indicates spread over years. For a similar depiction of precipitation see Figure B7b (b) Maps, averaged over the same time frame.

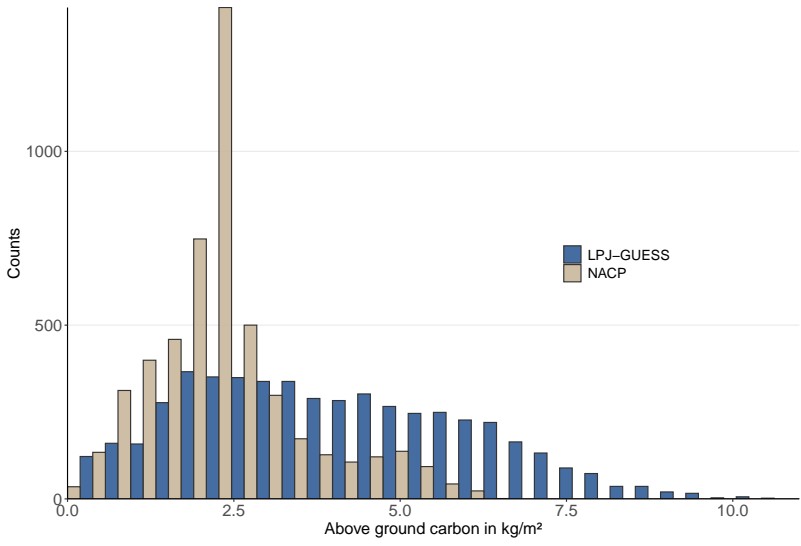

**Figure B2.** Comparing aboveground carbon (AGC) simulated by LPJ-GUESS with remotely-sensed AGC from NASA's NACP mission. Both datasets show the year 2005

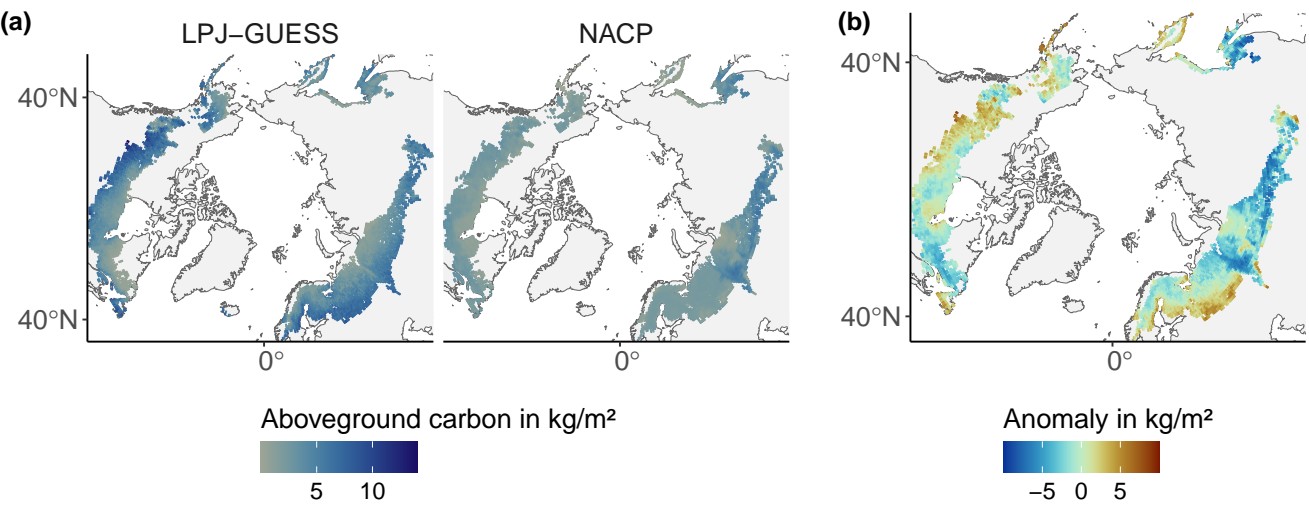

**Figure B3.** Comparing aboveground carbon (AGC) simulated by LPJ-GUESS with remotely-sensed AGC from NASA's NACP mission. (a) Absolute values (b) Anomalies. Both datasets show the year 2005 across the spatial domain.

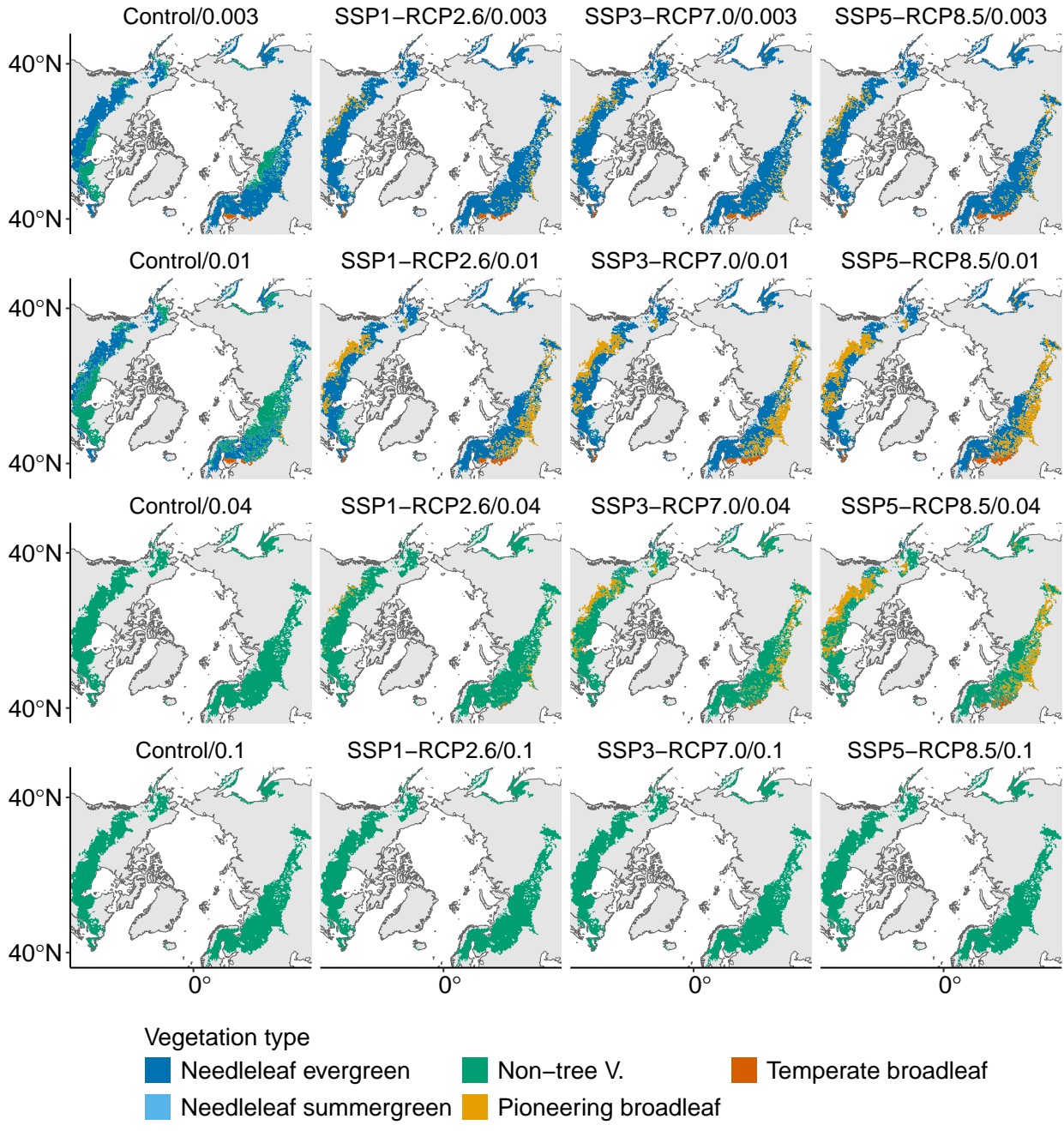

**Figure B4.** End-of-century dominant vegetation type for all configurations. Rows show a shift in climate, and columns a shift in disturbance regime.

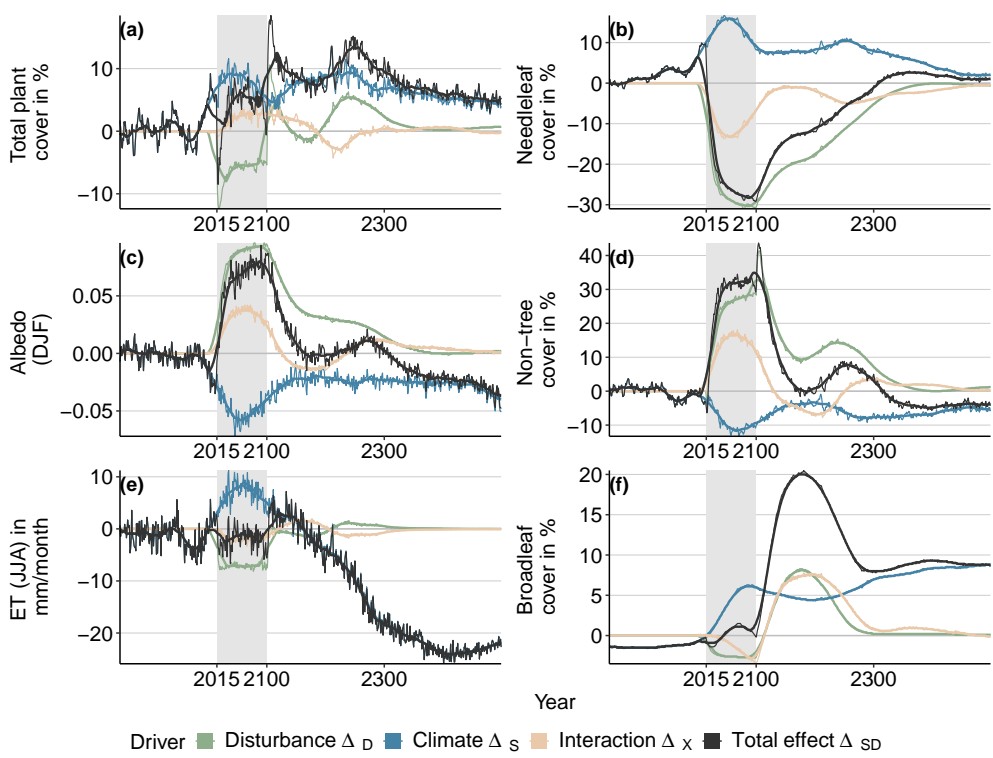

**Figure B5.** Total effects relative to control conditions and their attribution to different factors for vegetation composition, albedo, and evapotranspiration for the model configuration SSP1-RCP2.6/0.1. The black line indicates the difference between baseline (Control/0.003) and combined scenario SSP1-RCP2.6/0.1. The blue line difference between baseline and SSP1-RCP2.6/0.03, the green line difference between baseline and Control/0.1. The pink line gives the interaction effect (Difference between the black line and the sum of the green and the blue line). Grey boxes indicate the experimental period during which the disturbance regime is changed.

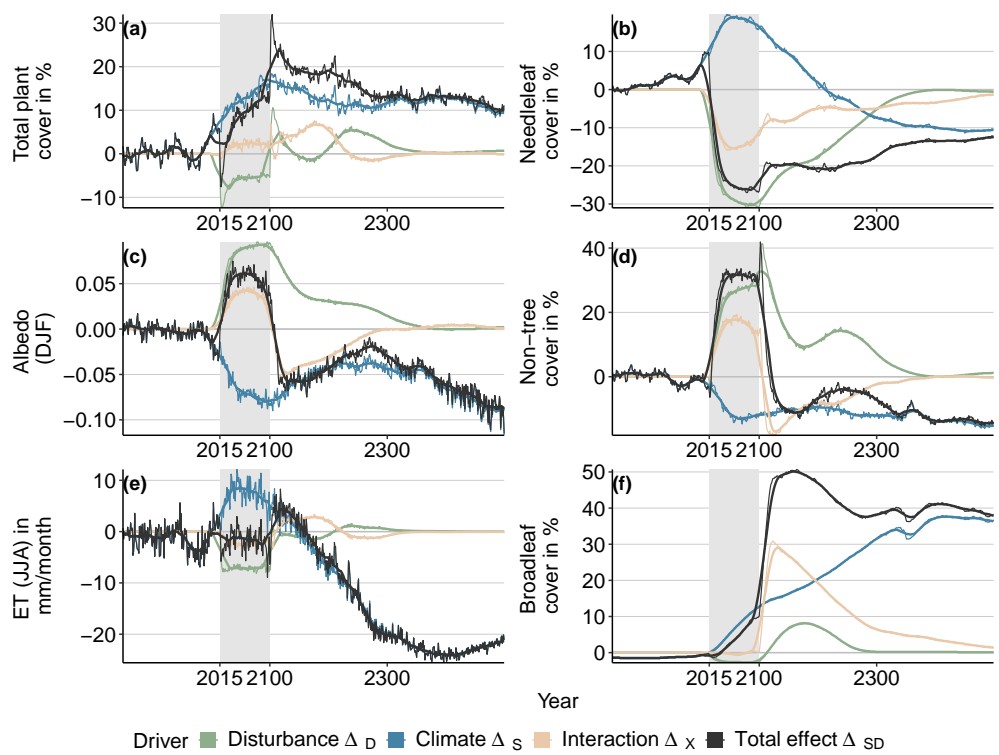

**Figure B6.** Total effects relative to control conditions and their attribution to different factors for vegetation composition, albedo, and evapotranspiration for the model configuration SSP5-RCP8.5/0.1. The black line indicates the difference between baseline (Control/0.003) and combined scenario SSP5-RCP8.5/0.1. The blue line difference between baseline and SSP5-RCP8.5/0.03, the green line difference between baseline and Control/0.1. The pink line gives the interaction effect (Difference between the black line and the sum of the green and the blue line). Grey boxes indicate the experimental period during which the disturbance regime is changed.

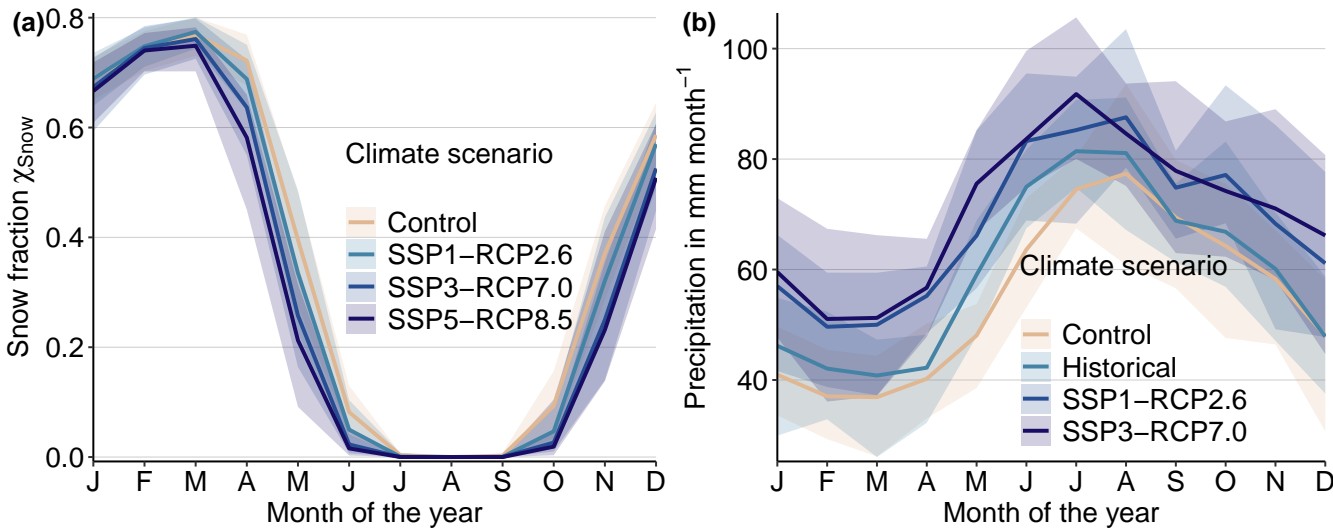

**Figure B7.** End of century seasonal curves of snow cover fraction (a) and precipitation (b). Thick lines indicate mean over 2070 - 2100, ribbons range over individual years.

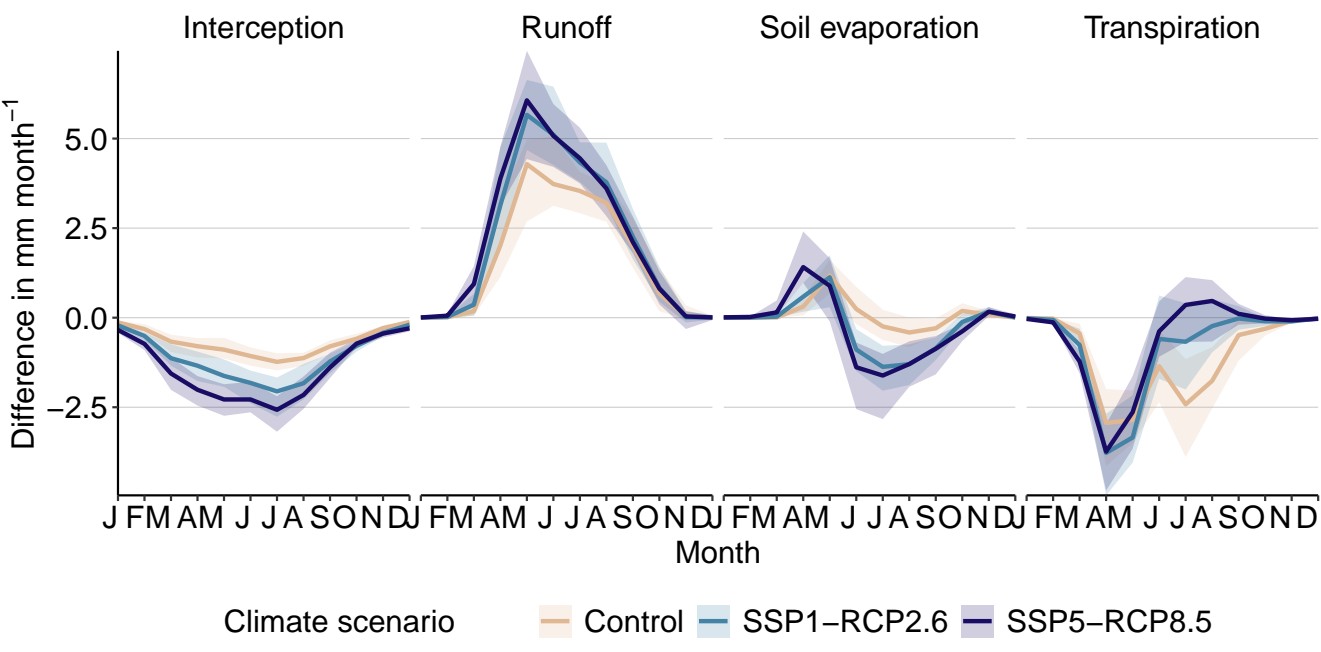

**Figure B8.** End-of-century seasonal curves of difference in state $\Delta_D$ between $p_D$ of $0.00\overline{3}$ and 0.04 for different evapotranspiration components as well as surface runoff. Evapotranspiration is the sum of interception, soil evaporation, and transpiration. Ribbon indicated spread of the year 2070 - 2100