# Peer review of "Disentangling future effects of climate change and forest disturbance on vegetation composition and land-surface properties of the boreal forest"

_EGUsphere, 2024_

## Author Response (AR1)

**Reviewer 1**

*We thank the reviewer for taking the time to go through the manuscript and providing constructive feedback. We are happy to hear the reviewer enjoyed reading the manuscript and have provided answers to the specific comments below.*

**Specific comments:**

**P3 – Line 74: "has been validated in previous studies ". Without going into too much detail, it would still be interesting to explain here how these validations were carried out, in order to ensure from the outset that the reader is confident about the validity of using this model.**

*Yes, that is a good point. We added more example studies and briefly outlined datasets and methods they used*

**P4 – Line 92: Where is Appendix A1 for a detailed description?**

*This is a legacy from an earlier version, thank you for spotting that. Establishment and mortality processes are not central to the discussion in the current version of the manuscript anymore, so we removed this Appendix to reduce the length of the manuscript. We removed the reference but added some more information on both processes in the Methods sections.*

**P4 – Line 93: Add a capital letter at the beginning of a sentence**

*Thanks for spotting that. We reworked this section due to a later comment.*

**P4 – Line 100: "From within the ISIMIP ensemble, data of the MRIESM2.0 Earth system model was chosen, as its response best represents the ensemble average." Based on what? In my opinion, taking just one climate model does not allow us to correctly determine the uncertainties of simulations. It would have been better to use two or three. What temporal resolution?**

*We refer to the medium climate sensitivity of MRIESM2.0 which results in an average temperature response within the ensemble(see https://www.isimip.org/protocol/isimip3b-temperature-thresholds-and-time-slices/ for a visualization and discussion). We adapted this in the text. The temporal resolution is daily (This is specified in the text). We agree that the use of multiple climate models would be ideal. However, since we are already performing a high number individual simulations due to our factorial experiments, we restrict to one climate model in order to reduce the computational costs.*

**Page 4 – Line 103: "Figure 1a shows the CO2 concentration, and Fig. A1 the climate data used. » Please explain a little bit more the projected changes in climate variables.**

*We added a paragraph to describe the used climate data in more detail and added more visualization of the data in the appendix.*

**Page 4 – Line 109: In table 1, you need to add more details here about the projected changes in climate variables, not only for temperature and not only for one period.**

*We added more information about how temperature develops over the scenario. More information on all climate variables is found in the appendix.*

**Page 4 – line 110: "on disturbance probability ». It is more appropriate to use "frequency" instead of probability.**

*Disturbances in LPJ-GUESS are modeled as a stochastic process. When using 'probability', we refer to the LPJ-GUESS parameter $p_D$, which encodes the disturbance probability. We agree that 'frequency' is more accurate when talking about observations.  We adapted this in the Method section to be more clear and made sure we use the appropriate terminology throughout.*

**Page 6 – Line 141: For what year? What area? And how?**

*We made our approach more explicit in the manuscript. We compare aboveground biomass for the year 2005 (year of the dataset) across the study area.*

**Page 9 – Line 185: "The range of total AGC compares well to satellite-derived data (Fig. B2), however, observed values are slightly lower and feature a pronounced peak around 3 kg/m2 not present in the modeled data." The peaks around three still represent a good proportion of the pixels... Can you show a map of anomalies between simulated and observed data to see where the model is simulating correctly?**

*We have added maps of modeled and observed carbon as well as the anomalies to the Appendix and extended the Results and Discussion accordingly.*

**Page 12 – Line 253: Add space between "October" and "(Fig. 4a)"**

*Changed, thank you for spotting this*

**Page 12 – Line 272: Remove space after Fig. 4a**

*Changed as well, thanks.*

**Page 12 – Line 274: Remove capital letter to Fig. 4a and B.**

*Changed as well.*

**Page 14 – Line 313: "previous modeling studies performed with LPJ-GUESS". What about other modeling studies performed with other models?**

*The studies referenced here conducted more extensive validations of LPJ-GUESS specifically within the Boreal zone, which is why we highlight that our results align with theirs, in the context of model validation. We discuss  studies conducted with other models e.g. in lines 47 – 54 of the manuscript.*

**Page 14 – Line 314: "and the range of total AGC corresponds to observations (Fig. B2). »
I'm not sure if you sum up all the ABC shows on Fig. B2.**

*Unfortunately, we are not entirely sure what is meant here. We hope our extended discussion of
the validation data has made things clearer.*

**Page 14 – Line 315: Scandinavia**

*Changed, thanks for spotting.*

**Page 19 – Line 415 : « It should be noted that ecophysiological control on ET is relatively
weak in the current version of LPJ-GUESS. » I agree, so why study it if we can't really
distinguish causes of changes?**

*This is the currently published version of LPJ-GUESS, and therefore, we want to show the
sensitivity of the current model version to disturbance effects and their interaction with climate.
We do not intend to make statements about the most likely future outcome. With this statement
we wanted to highlight, that current model developments of plant hydraulics might affect the
results. We rephrased to the following: 'It should be noted that some processes governing
ecophysiological control on ET are lacking in LPJ-GUESS. Yet, in global simulations LPJ-
GUESS has been shown to accurately simulate ET compared to a number of hydrological
datasets (Zhou et al 2024). However, future model development aiming for a better
representation of plant hydraulic strategies may be able to further improve model
performance.'*

**Page 20 – Line 438: "This means that regeneration failure - eroding of seed banks
through too frequent disturbances (Turner et al. (2019); Hansen et al. (2018)) - cannot
occur by model design. Here, different reproductive strategies, e.g. serotony or
resprouting, would also be important to consider". The fact that regeneration accidents
are not included is a significant limitation that needs to be taken into account in order to
strongly balance the results of this study. See studies on this subject in Quebec (e.g.
Girard et al. 2008)**

*Yes, we agree. We made it clearer in the discussion and the conclusion that this is a severe
limitation to interpreting recovery potential.*

**Page 20 – Line 443: "However, it is important to keep in mind that specific disturbances
can have additional effects we are not considering here." Yes additional effect and to go
further, you need to discuss about amplifying effects between disturbances.**

*This is a good point, we added it to the discussion.*

**Page 20 – Line 445: "Further, our disturbance dynamics are neither linked to climate nor
to vegetation type, as, for example, a detailed fire disturbance module would be."
SPITFIRE is already implemented in LPJ so why you didn't used it?**

*We opted for the simple disturbance module instead of a dynamic fire module like SPITFIRE or
Blaze or a number of reasons:*

- *We are interested in the overall disturbance regime, not just fire. Equally detailed modules do not exist, for, e.g., spruce budworm, bark beetles, or windthrow, in LPJ-GUESS yet.*
- *Future projection of process—based disturbance modules are associated with large uncertainties. Our stochastic framework makes these explicit while being parsimonious*
- *We were specifically interested in the interaction effect between climate change and disturbances, requiring factorial simulations that are decoupled by design. What we meant in the above phrase is that disturbances are uniform on space and time. We refined this in the manuscript*
- *Using a process-based fire module adds further dimensions, e.g. burn severity or burned area that we would have had to accommodate in the factorial experiments. We here wanted to apply a parsimonious approach but agree that a similar study with a process-based model would produce further insights.*

*We hope we could clarify the points raised. Thank you again for taking the time!*
* * *
**Reviewer 2**

**In this paper, the authors disentangle the influence of climate and disturbance on future changes in vegetation cover. Using LPJ-GUESS, the authors conduct multiple experiments to test the sensitivity of future boreal forest vegetation changes to climate change and disturbance. Additionally, the authors also extend their simulations for multiple centuries to observe transient regional changes in vegetation cover if disturbances were reduced. The authors find that warmer climate scenarios lead to more vegetation cover and aboveground carbon while disturbance decrease aboveground carbon and significantly change the distribution of vegetation. Additionally, these changes in vegetation affect albedo and evapotranspiration. The authors highlight the importance of climate and in combination the changing disturbance regime when considering future changes in boreal forests.**

**A major strength of the manuscript is the use of multi-century simulations and their experimental set-up to tease apart the influence of changing climate and disturbance regimes on the transient nature of boreal vegetation dynamics. A weakness of the manuscript is that disturbance itself changes with climate and vegetation composition which the model is unable to simulate limiting the authors to focus more on the influence of disturbance frequency. Overall, this reviewer found the study to be interesting and believes that the paper should be considered for publication after addressing comments and questions.**

*First of all, we thank the reviewer for taking the time to read the manuscript and for the extensive and thoughtful feedback. We are happy to hear that the reviewer found the study interesting and will in the following address the comments and questions made.*

- **The authors only analyze one post-2100 simulation. This reviewer is left wondering what the other simulations look like and how regeneration varies depending on the 2100 ecosystem state. Currently by setting the return interval to roughly 300 years from 25 years or so, one would expect disturbance to have less influence over a couple centuries since that is what the fire return interval determines. However, part of the strength of this study and what makes it unique is having multi-century simulations. With these simulations the authors can compare the differences in the spin-down amongst the simulation and gain additional insight helping answer their questions. For example, SSP5-RCP8.5/0.04, simulation of focus, effectively becomes RCP8.5/0.003 for the spin-down. The authors then could compare SSP58.5/0.04 to RCP8.5/0.003 and see the differences between the simulations for the years 2100-2500 as they approach steady state. Do the authors expect to see all RCP8.5 scenarios to converge by 2500? How do different ecosystem states influence the post-2100 long-term changes.**

*Yes, this is a good point. We took a closer look at other scenarios and will discuss this, but would like to add two remarks first :*

*1. The reason we set the disturbance interval back to baseline after 2100 is to investigate the reversibility of disturbance-induced impacts. We considered performing a second set of long-term simulations without reseting the disturbance intervals, but decided against it due to computational capacities and to no overload the study. We agree that this would be an interesting set-up as well.*

*2. Figure 3 does not only analyze one post-2100 simulation. It shows the difference between Baseline (Control/0.003) and Control/0.04 (green line), Baseline and SSP585/0.003 (blue line ), Baseline and SSP585/0.04 (black line) as well as the residue between the black line and the green + blue line (Interaction effect, pink/rose line). We updated Figure 1 (Methods) to hopefully make this clearer. Regarding the reviewers question 'The authors then could compare SSP58.5/0.04 to RCP8.5/0.003 and see the differences between the simulations for the years 2100-2500 as they approach steady state. Do the authors expect to see all RCP8.5 scenarios to converge by 2500?' Comparing SSP58.5/0.04 to RCP8.5/0.003 is comparing the black and the blue lines in Figure 3, and yes indeed, these converge in all cases and also for other disturbance probabilities.*

*We nevertheless agree that it would be worthwhile so analyze more scenarios in the same way. Since looking at al combinations in detail would overload the paper, we added the same analysis for the configuration SSP1RCP2.6/0.1  and SSP5-RCP8.5/0.1to the Appendix and discuss it in the paper.*

- **This reviewer believes that the manuscript can be improved if the authors consider restructuring some of the paragraphs either by merging or elaborating more on paragraphs and sentences to help with presentation of the story. For example, in the methods there are some sections that are just two sentences. The authors are at times vague when describing methods and their results and**

**providing more information will help improve the clarity of the manuscript. The authors should also be thorough to make sure that all tables are figures are formatted correctly and their figures, tables, figure captions, and text do not have errors in them. There are many instances throughout the text where figures and their descriptions do not match. In the discussion section please also consider referencing the figures and tables more frequently so readers can find what the authors are discussing.**

*We restructured the Methods section, revised the referencing of Figures and Tables. We kept this paragraph in mind when addressing specific comments, especially to make sure the Methods section more to the point.*

**Specific Comments:**

**P2 L20-24: Johnstone et al. 2016 might be a good reference to cite since they talk about ecosystem states and stability. This paragraph might also be rewritten to better explain disturbance regimes and how that connects to the change in disturbance regimes that lead to trends in forest cover loss.**

**Johnstone, J. F., Allen, C. D., Franklin, J. F., Frelich, L. E., Harvey, B. J., Higuera, P. E., Mack, M. C., Meentemeyer, R. K., Metz, M. R., Perry, G. L., Schoennagel, T., & Turner, M. G. (2016). Changing disturbance regimes, ecological memory, and forest resilience.** *Frontiers in Ecology and the Environment*, *14*(7), 369–378. https://doi.org/10.1002/fee.1311

*We agree, this is a very relevant paper, we added the reference and slightly expanded the paragraph.*

**P2 L35-40: The authors might consider including Randerson et al. 2006 here or another section given that the author's study is studying the impact of fires on boreal forest dynamics and the effects of climate.**

**Randerson, J. T., Liu, H., Flanner, M. G., Chambers, S. D., Jin, Y., Hess, P. G., Pfister, G., Mack, M. C., Treseder, K. K., Welp, L. R., Chapin, F. S., Harden, J. W., Goulden, M. L., Lyons, E., Neff, J. C., Schuur, E. A. G., & Zender, C. S. (2006). The Impact of Boreal Forest Fire on Climate Warming.** *Science*, *314*(5802), 1130–1132. https://doi.org/10.1126/science.1132075

*Yes, agreed, another important study. We added a reference here.*

**P2 L41-46: At this point in the manuscript, the question(s) that the authors are interested or are referring to have not been clearly stated. The authors could rearrange the sentence or state the broad question regarding boreal vegetation dynamics. Also, it is unclear what remains incomplete. Regarding observational studies, space-for-time chronosequences have been used to study the long-term effects on ecosystems that go beyond 30-years where successional trajectories can be estimated for more than a century. The authors might have a sentence or two on space-for-time analysis and why it is still difficult to pinpoint the changes.**

*Good point. We restructured the paragraph to make both aspects clearer.*

**P3 L 58: In this study, we use the DVM LPJ-GUESS (Smith et al. (2001, 2014)) to fill this gap. Since this is the first sentence of the paragraph, the authors might want to be clear about what this gap is referring to.**

*We believe this is now clearer through addressing the previous and next comment.*

**P2-3 L54-57:**

**"Additionally, none of these studies explored long-term effects and regeneration after disturbance. To our knowledge, there so far exist no modeling studies that systematically investigated the immediate and long-term (centennial) impacts of both changing disturbance regimes, climate change, and their interaction on evergreen boreal forest dynamics across the biome for different climate futures."**

**This reviewer believes that the authors should take care with a statement like this. For example, there are Holocene studies where the idea is to use the Holocene as a proxy for different climate futures which can look at changes similar to short term and long term changes climate and disturbances. These studies use dynamic vegetation models and explore the role of disturbances and/or different climate conditions on boreal forest dynamics (Chen et al. 2019, Chen et al. 2022). There are also other modeling studies that investigate boreal dynamics in Alaska using different SSP climate scenarios are considered as well as analysis of the changing disturbance regimes given that vegetation itself feedback on the disturbance regime (Foster et al. 2019, Foster et al. 2022). It is also unclear what is meant by short-term vs long-term and why the authors use these definitions given no explanation of the timing of successional trajectories or fire return intervals in the introduction. The authors could provide additional references or introduction on the fire return intervals and temporal scale of successional trajectories and what short-term vs long-term mean in the context of this study. This reviewer believes that the authors might want to reconsider how these sentences are written and try to simplify and explain how the model builds upon previous work.**

**Chen, W., Zhu, D., Ciais, P., Huang, C., Viovy, N., & Kageyama, M. (2019). Response of vegetation cover to CO2 and climate changes between Last Glacial Maximum and pre-industrial period in a dynamic global vegetation model. *Quaternary Science Reviews*, *218*, 293–305. https://doi.org/10.1016/j.quascirev.2019.06.003**

**Chen, J., Zhang, Q., Kjellström, E., Lu, Z., & Chen, F. (2022). The Contribution of Vegetation-Climate Feedback and Resultant Sea Ice Loss to Amplified Arctic Warming During the Mid-Holocene. *Geophysical Research Letters*, *49*(18), e2022GL098816. https://doi.org/10.1029/2022GL098816**

**Foster, A. C., Armstrong, A. H., Shuman, J. K., Shugart, H. H., Rogers, B. M., Mack, M. C., Goetz, S. J., & Ranson, K. J. (2019). Importance of tree- and species-level interactions with wildfire, climate, and soils in interior Alaska: Implications for forest change under a**

warming climate. *Ecological Modelling*, *409*, 108765. https://doi.org/10.1016/j.ecolmodel.2019.108765

Foster, A. C., Shuman, J. K., Rogers, B. M., Walker, X. J., Mack, M. C., Bourgeau-Chavez, L. L., Veraverbeke, S., & Goetz, S. J. (2022). Bottom-up drivers of future fire regimes in western boreal North America. *Environmental Research Letters*, *17*(2), 025006. https://doi.org/10.1088/1748-9326/ac4c1e

*The you for the comment and the literature recommendations. We reformulated this paragraph to make sure that the gap we identify are biome-wide 21st-century as well as long-term future projections (beyond the 21st century) that investigate changes in both climate and disturbance regimes.*

**P3-L73: Appendix A seems to have a formatting issue since Table A2 is in appendix B. The tables should be ordered by appearance in the text. A1 is referenced after A2. The current table A2 should then be A1 instead.**

*Thank you for spotting that. We fixed it.*

**P4-L110-113: This information could also be included in the intro to help readers understand what the immediate vs. long-term definitions are.**

*This is a good point, we added it to the end of the introduction*

**P5 L118-126: Can the authors explain what they mean when they recycled data from 2095-2100? Please be specific about what is being recycled and what is not for the post-2100 simulation. Also, how are the authors analyzing the post 2100 period?**

*For the years 2101 – 2500, we created time-series of temperature, precipitation and radiations by randomly sampling the data of the years 2095 – 2100 to create constant time-series with some inter-annual variability. We used CO2 from the year 2100 for the whole period. We clarified this in the methods as well.*

**P6 L135: The authors might reconsider calling shrubs and grasses the tundra. Tundra is a particular ecosystem that is more than just an area containing shrubs and grasses. On P4 L114-117, the authors also mention they use the Taiga ecoregion from the WWF ecoregion dataset which is different than the tundra ecoregions.**

*Yes, this is a good point, we agree with the reviewer that our use of the term 'Tundra' here is not accurate. We adapted the manuscript to distinguish between Tundra ecosystems and non-tree vegetation within the Boreal forest zone.*

**P6 L139-140: Are these area weighted averages, are you including areas with water etc.? It may help to reiterate in your data analysis section that you are focusing on the WWF taiga ecoregions.**

*Thank you for the question. Large waterbodies are excluded from the simulations. We made this clearer in the manuscript.*

**P6 L141-143: Is the biomass dataset from NASA ABoVE? It does not appear in the ABoVE ORNL DAAC when searching for biomass https://daac.ornl.gov/cgi-bin/dataset_lister.pl? p=34. There are ABoVE biomass datasets that the authors could compare their results to as well if they so choose. As mentioned in a later comment please provide references to datasets used in the study for readers to find the datasets more easily. What years are the authors comparing their model output?**

*Thank you for the question, this was a mistake from our side. The dataset we are comparing to is indeed not from AboVE but from the NCAP mission. We made our approach clearer in the text. We cited the datasets as instructed on DAAC ORNL: https://daac.ornl.gov/VEGETATION/guides/Eurasia_Biomass.html and https://daac.ornl.gov/NACP/guides/NACP_Boreal_Biome_Biomass.html*

**P7 L145: Which wavelengths is the albedo measurement for?**

*LPJ-GUESS does not output albedo, this is calculated based on the specific albedos from Boisier et al. From their methods : 'The datasets used in this study gather satellite products and global simulations from the LUCID model inter-comparison project (Table 1). The MODIS shortwave broadband bi-hemispherical reflectance (white-sky albedo) and snow cover data (MCD43C3; Schaaf et al., 2002; Gao et al., 2005), in addition to the land cover product (MCD12Q1; Friedl et al., 2010), were used to derive snow-free and snow-covered albedos of different land cover types'*

**P8 L181-186: At what point after the spin-up period are the authors getting their values from? Please be clear on what years are being referenced. The authors mention range, and difference in peak of the distribution from their model validation. Does this mean that the model overestimates biomass regionally? Can the authors please provide some statistics around their validation?**

*We added some information and analysis here. Yes, the model does overestimate biomass in some regions due to the low disturbance return intervall and the absence of  forest management. We discuss this in the manuscript. The spin-up period ends in 1850.*

**P8-9 L182-185: The sentence is confusing because it could be read as 59% because of the excluding soil portion. Please make it clear that you are looking at the absolute coverage and not percentage when excluding soils**

*Unfortunately we are not completely sure what this comment is referring to but take that we have not been entirely clear with the metrics we are using:*

*Fractional plant cover (FPC) as outputted by the model gives the fraction of ground covered by a PFT. If total vegetation FPC > 1 we have dense, multi-layered vegetation. If total vegetation FPC < 1, we have sparse vegetation and we can calculate the fraction of bare soil as 1 – total vegetation FPC. When we talk about vegetation composition, we are interested in the contribution of a given PFT to vegetation cover, that is the total vegetation FPC (excluding soil). So the 59% here means that 59% of the vegetation cover is comprised of needleleaf evergreen trees. We report vegetation composition in this manner, as there is a large variability in*

*vegetation density throughout the study region and absolute FPC values are a less meaningful to describe changes in vegetation composition.*

*We adapted the methods section to define the metrics we use more clearly and adapted throughout the manuscript.*

**P9 L186: The authors should keep the format of units the same as the rest of the manuscript kg/m2 or kg m^-2. Please keep a space between the units as well.**

*Yes, thank you for spotting that. We made sure we use kg m$^{-2}$ throughout the text*

**P9 L195: For figure 2 it would help for each map in panel c) to have their own panel label.**

*We agree. I spend some time on it but unfortunately this is a non-trivial thing to achieve here without manually altering the plot, so we omitted this. We hope this is fine.*

**P9 L202: The authors might consider using the fire return interval value here instead of the probability of disturbance to better provide context on the frequency of fires.**

*Unfortunately we do not understand this comment here. We do not simulate fire separately, so frequency of fires would not make sense in this context.*

**P10 L223: Remove 3**

*This was meant as a reference to Figure 3, we corrected it.*

**P11 L229: Figure 3 might need an update since the authors refer to a pink line, but none is observed. The panels are also not labeled, but the authors refer to panel a).**

*Yes, this figure used to look different, thank you for spotting that. We adapted the caption. There should be a pink line visible. Maybe this is a language issue? We made the colors of the plot more vibrant to make sure they are well visible.*

**P12 Figure 3: Please give each plot a panel label. For clarity, please describe what each line and their color represent. This reviewer thinks that the unit for the percent on panels 1-4 (a-d) is actually in fraction rather than percentage. The fraction vs. percent issue is present for figure 2 as well.**

*Yes, we think the reviewer is correct about the percentages. We corrected this in the plot, added labels and extended the figure caption.*

**P13 L266: Would disturbance frequency be more appropriate since the intensity of disturbances in the model is constant?**

*Yes, we see the point. We used 'intensity' here mostly to have more variety of language but 'intensity' and 'frequency' might not be synonyms. To clarify : we use 'frequency' when referring to observation and 'probability' when referring to the LPJ-GUESS parameter $p_D$.*

**P15 Figure 4: Are these figures the most up to date figures? I do not see any stippling. Should it be p > 0.01 since those are what are being stippled.**

*There was an issue with the figure rendering which caused the stipples to not be visible in some pdf readers and when printing the paper. We replotted the figure to address this issue. We stippled the areas with non-significant changes, as due to the size of the grid cell, the fill color of the cell is not properly visible otherwise.*

**P16 Figure5: Same comment as before.**

*See above.*

**P17 L346-347: There -> their. Double elevated**

*Thanks for spotting that. We corrected it.*

**P17 L348-349: The absolute abundance for summergreen goes to near zero or is this in reference to just fractional cover? Seems like it doesn't change their relative abundance but absolute does since absolute abundance for all vegetation decreases with more disturbance.**

*Yes, under control climate summergreen trees are not very abundant and we would argue that an increase in disturbance alone does not really change that. For the warming climate scenarios we agree, their higher abundance with higher disturbance seems to be related to their higher resilience ot disturbances.*

**P20-21 L425-455: Here is where the additional analysis on the differences between transient might improve the discussion for section 4.3. The authors might want to elaborate how these mechanisms potentially affect the simulation results beyond that the model overestimates.**

*Thank you for the feedback. We expanded this section, referencing additional long-term scenarios.*

**P21 L475-480: Please put the reference and url for the datasets used in this study. For example, the authors mention using aboveground biomass data for validation from Neigh et al. (2013,2015). The authors should also include the input data for their models as well.**

*The datasets used for validation ere Neigh, 2013, 215 and Margolis, 2015. The input data is Lange, 2021. The DOIs in the list of reference link to the original datasets.*

**P31 Table A2: Please include what each of the variables are in the description.**

*Good point, thank you. Added.*

**P34 Figure B3: The authors might consider using this figure in the main figure set. This was interesting to look at and helps put the histogram bars into a spatial context.**

*Thank you. We decided to leave this in the Appendix as it is quite a clunky figure, but made sure we referenced it more often in the main text.*

*He hope we were able to address all comments thoroughly and again thank the reviewer for taking the time to provide this helpful feedback.*

---

## Author Response (AR2)

**Detailed point-by-point response**

Overall, this reviewer found that the manuscript has improved and the authors have addressed many concerns. There are some minor things that this reviewer would like some clarification on. Additionally there are some technical corrections that the authors need to address as well. Primarily around the use of defined terms, figure corrections, etc. Please be extremely thorough with edits around formatting and how you decide to use your defined variables. This review is based on the manuscript version with changes shown, so some of these comments may not be relevant and already addressed.

*We thank the reviewer for taking the time to re-read the manuscript and to provide feedback on the manuscript and appreciate their thoughful and diligent comments. We hope we have adressed them thoroughly and ask the reviwer to get back in case something is not clear or unsatisfactory answered.*

**Specific Comments:**

1. L26 Pan et al. 2024 might be a good reference as an update: Pan, Y., Birdsey, R. A., Phillips, O. L., Houghton, R. A., Fang, J., Kauppi, P. E., Keith, H., Kurz, W. A., Ito, A., Lewis, S. L., Nabuurs, G.-J., Shvidenko, A., Hashimoto, S., Lerink, B., Schepaschenko, D., Castanho, A., & Murdiyarso, D. (2024). The enduring world forest carbon sink. Nature, 631(8021), 563–569. https://doi.org/10.1038/s41586-024-07602-x

   *Yes, that's a good point. Included!*

2. L93 maybe include resolution for the study here as well? I see that it's in L112, but one of the first things that popped into my head was the question of how big are the grid cells.

   *Agreed, we included the resolution in L93.*

3. L134-137 Could the authors please plot out the map of the the ecoregions alongside the evergreen needleleaf fraction map to show clearly how the study/model areas are defined? Could be added to the supplement. On all maps maybe have some clear way to distinguish the study area because otherwise the maps such as figure 2 make it look like everything is bare soil even though the grey is slightly different than the background grey.

   *We adapted the color key of bare soil in Figure 2, to be better distinguishable from the base map. We also added the WWF boreal ecoregion to our Figure 3B for reference. If the reviewer is interested further, here is an interactive map of all the ecoregions: https://ecoregions.appspot.com/.*

4. Figure 1 setpup → setup, really nice figure!

   *Corrected. Thanks for the feedback, really appreciated!*

5. L152-160: some formatting issues, where equation showing up after Table 1. I like the inclusion of the equations. Maybe reference the equations in the description. This is also the area where this reader got confused with the terms and how they are used throughout the manuscript.

   *We expanded on the definitions given here to hopefully clear out confusing use of these variables throughout the manuscript (see later comments). We also corrected some formatting issues in the process and made sure the table was not inserted right before th equation (though we assume this will be subject to final layout changes).*

6. "To express which percentage of vegetation FPC(excluding bare soil) consists of which PFT". Is soil specific in the model of land where vegetation can grow or is just land fraction/cover? Could the authors consider changing soil to land as a general term and then specify bare soil as you have already done to make a clearer distinction. For example "FPC describes the fraction of (soil → land), covered by a specific PFT. If FPC is smaller than one, vegetation does not cover the (soil → land) completely, and the bare soil fraction is calculated as 1 - FPCV ".

   *This might be due to the fact that we use a soil map to define our grid and I am therefore thinking of the landsurface as 'soil' but 'land' sounds not correct to me here. I would understand this to refer to larger geographic areas and not the per-gridcell level. But I am happy to change it, if the reviewer feels strongly about this.*

7. L163-165: What instances are IBS and TeBS combined? Region based?

   *We combined this for Figure 3. It's a good point, I made it explicit in the text.*

8. L180 is bare soil 1-FPCv and not $X^V$

   *The difference between $FPC_V$ and $X^V$ is that $FPC_V$ can be larger than one, while $X^V$ and $X^{Soil}$ are normalized between 0 and 1 (due to being fractions). Equation 4 says that $X^{Soil}$ is 0 if $FPC_V > 1$ (no bare soil), or $1 - FPC_V$ if $FPC_V < 1$, so make sure it is a fraction. Now thinking about it, $X^V$ really is a synonym for $FPC_V$ that is only defined if $FPC_V < 1$. We removed $X^V$ throughout the text and used $FPC_V$ to avoid confusion here.*

9. L187 Is the heading 2.4.3 still there?

   *One reviewer requested less subsections, so I merged 2.4.2 and 2.4.3. The current headings are: 2.4.2. Biophysical land-surface properties and 2.4.3. Attribution of drivers*

10. Table B1 units for FPC $m^2 \, m^2$?

    *Yes, thank you. Corrected.*

11. L216 Is it FPCv or $X^v$? What exactly is $X^v$ without the $Xi^v$

    *See answer to 8. I also added some information to the paragraph to hopefully make it clearer.*

12. L220 Potentially change units for AGC to kg C $m^-2$.

    *Yes, it should be kg C $m^-2$.*

13. Figure 2 2070-2100. Is there really that much bare soil fraction? Can the authors explain this further?

    *In the model, yes. One aspect is that these are averages over the whole study area and parts of the study area are very high latitudes with very sparse vegetation. Another thing to keep in mind is that these fractions are resolved in 'real space' whereas remote sensing data classifies on a pixel basis e.g. 50 % forest cover is classified as a forested pixel in the ABoVE framework. Thus vegetation cover fractions tend to be higher in remote sensing. We have another study where we compare and validate this in more detail, unfortunately it is not published yet.*

    *It is of course possible that our simulated bare soil fraction is indeed to high. This would especially have an impact on the albedo calculations. I added a sentence to the discussion there to make this clear.*

14. L253 "of total FPCv" I think if the authors define terms that it would be best to use them or consider ways to simplify the terms.

    *Total FPC includes soil fraction, so $FPC_V$ would not be correct here. I made it clearer in the text, which variable I am refering to where and also references the Equations that are now Eq. 1 - 4, previously Eq. 1.*

15. L269 "$X^v$ -¿ FPCv?" declines?

    *I think this also stems from the confusion due to $FPC_V$ and $\chi^V$ being synonyms if $FPC_V < 1$, as is the case for averages over the landscape (that refers to Figure 2a and Figure 3). I initially used $\chi^V$ here for consistency but I agree that I might have achieved to be more confusing instead. I changed it here to $FPC_V$. See also answer to 8.*

16. L270 Figure 3b

    *We added 'Figure', thanks for spotting that.*

17. L281 Figure ?? formatting issue.

    *Yes, thank you. Corrected.*

18. L296 is $X^v$ this seems like the authors are trying to state that FPCv increases?

    *See Comment 17.*

19. Figure 3, B5, and B6 are awesome and interesting to think about. For easier comparison these figures should be kept on the same scale.

*Thanks, I am happy to hear that. Good point, I set the y-limits to be consistent across plots. (I hope this is what you mean?)*

20. Figure 4 no pink lines or stippling

*Unfortunately I do not understand this comment Are you saying the pink lines and the stippling should not be there or that you do not see it? Would it be possible to clarify?*

*Thank you again for taking the time and apologies we were not able to adress the last comment.*